# Understanding Performance Collapse in Layer-Pruned Large Language Models via Decision Representation Transitions

Boyu Shi [1 2]  Chang Liu [1 2]  Chuanbao Gao [1 2]  Xu Yang[* 1 2]  Xin Geng[* 1 2]

## Abstract

Layer pruning efficiently reduces Large Language Model (LLM) computational costs but often triggers sudden performance collapse. Existing representation-based analyses struggle to explain this mechanism. We propose studying pruning through decision representation. Focusing on multiple-choice tasks, we introduce two metrics, Decision Margin and Option Frequency, and an Iterative Pruning method to analyze layer-wise decision dynamics. Our findings reveal a sharp decision transition that partitions the network into two stages: a Silent Phase, where the model cannot yet predict the correct answer, and a Decisive Phase, where the correct prediction emerges. We also find that pruning the Decisive Phase has minimal impact, whereas pruning the Silent Phase triggers immediate performance collapse, highlighting its extreme sensitivity to structural changes. These results indicate that disrupting the Silent Phase is a dominant structural mechanism behind pruning-induced collapse in the evaluated settings, because it prevents the critical decision transition from occurring. The code is available at https://github.com/shiboyu1999/IT-Prune.

## 1. Introduction

Large Language Models (LLMs) have demonstrated remarkable capabilities but pose significant deployment challenges due to their huge parameter scales (Liu et al.; Ashkboos et al., 2024; Shi et al., 2026a). Layer pruning (Men et al., 2025; Qiao et al., 2025; Wang et al., 2025) has emerged

---

* Co-corresponding authors. [1]School of Computer Science and Engineering, Southeast University, Nanjing, China [2]Key Laboratory of New Generation Artificial Intelligence Technology and Its Interdisciplinary Applications (Southeast University), Ministry of Education, China. Correspondence to: Xu Yang <xuyang_palm@seu.edu.cn>, Xin Geng <xgeng@seu.edu.cn>.

*Proceedings of the 43rd International Conference on Machine Learning*, Seoul, South Korea. PMLR 306, 2026. Copyright 2026 by the author(s).

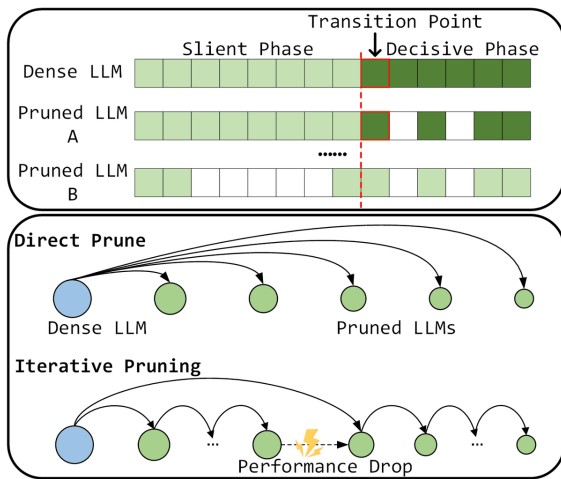

*Figure 1.* (Top) Conceptual illustration of the Silent Phase and Decisive Phase as pruning rate increases. Colored squares denote retained layers and white squares denote pruned layers; the red dashed line marks the dense model's transition point. (Bottom) Comparison between one-shot/direct pruning, which prunes the dense model directly to a target size, and Iterative Pruning (IP), which removes layers step by step, and restarts from a better state when a pruning step triggers a sharp performance drop.

as a compelling, training-free strategy to mitigate these computational costs. However, its practical utility is often constrained by a "cliff-edge" effect: performance remains relatively stable up to a specific threshold, beyond which it collapses abruptly (see Figure 2 (Top)). This highly non-linear degradation suggests that the model's functional integrity depends on specific structural properties that are not yet fully understood (Hu et al., 2025).

Existing literature (Gromov et al., 2024) primarily attributes this collapse to layer-wise functional localization, arguing that pruning disrupts critical foundational knowledge or higher-level reasoning modules. However, this representation-centric view is increasingly challenged by empirical evidence. As illustrated in Figure 2 (Bottom), our Centered Kernel Alignment (CKA) (Kornblith et al., 2019) analysis reveals a representation-performance conflict: even at a 50% pruning rate where task performance has entirely collapsed, the pruned model's hidden states retain surprisingly high semantic alignment with the original dense model across nearly all layers. This persistence of deep represen-

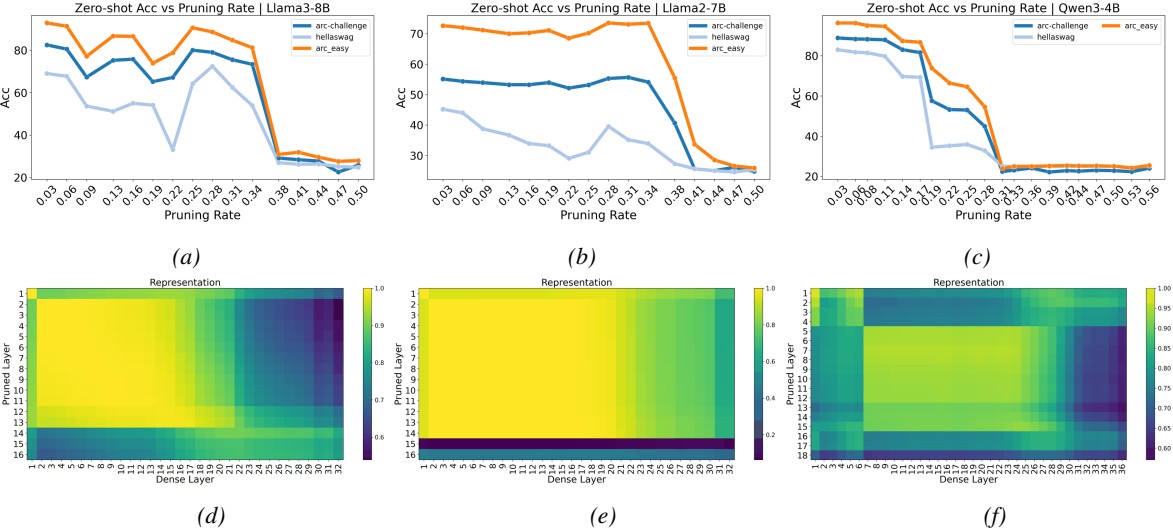

*Figure 2.* (Top) Zero-shot accuracy across various benchmarks as a function of pruning rate for Llama3-8B, Llama2-7B, and Qwen3-4B. (Bottom) CKA-based similarity heatmaps of hidden representations between dense and 50%-pruned models on the Hellaswag task. Despite sharp performance collapse, deep-layer semantic representations remain largely preserved, suggesting collapse is not driven by the loss of hidden features.

tational structures despite functional failure implies that internal feature similarity is a poor proxy for task-level competence. Consequently, a fundamental question arises: **If high-level hidden representations remain largely intact, what structural disruption drives the sudden collapse of the model's ultimate decision-making behavior?**

To answer this question, we shift our focus from hidden representations to a decision-centered perspective. Instead of looking at how information is encoded, we study how decisions actually emerge as data passes through the layers. We use multiple-choice tasks for this study because they have clear correct answers and a fixed set of options. This makes it much easier to track decision behavior than in open-ended generation, where semantic validity is difficult to quantify. We introduce **Decision Margin (DM)**, defined as the probability gap between the ground-truth option and the most likely alternative. Interestingly, we find that decision formation is not a gradual process. Instead, DM remains negative in the early layers and undergoes a sharp transition at a specific depth (called transition point), beyond which the model consistently favors the correct answer.

This abrupt transition naturally partitions the LLM into two functional stages: **a Silent Phase**, where the model is yet to identify the correct response, and **a Decisive Phase**, where reliable decision-making emerges, as in Figure 1 (Top). To further characterize these stages, we propose **Option Frequency (OF)** to measure the *layer-wise distribution* of predicted options. In the Silent Phase, we observe a severe distributional collapse, where the model's predictions are dominated by a single option regardless of the input. Conversely, the Decisive Phase exhibits a balanced distribution, indicating that the model has escaped its internal bias to form a structured decision.

Building on this framework, we investigate how layer pruning disrupts this delicate decision structure. Existing pruning methods typically remove blocks of layers simultaneously (Men et al., 2025; Wang et al., 2025), which conflates the effects across different phases and obscures the precise mechanics of performance collapse. To achieve a more granular inspection, we employ Iterative Pruning (IP), which is a greedy, layer-by-layer removal strategy. By minimizing structural perturbations in each step, IP serves as an analytical probe, allowing us to progressively track how instabilities propagate through the decision structure and pinpoint the exact moment the transition point is lost.

Our experiments across diverse LLMs (including LLaMA3-8B (Dubey et al., 2024), LLaMA2-7B (Touvron et al., 2023), and Qwen3-4B (Yang et al., 2025)) reveal a universal pruning trajectory: the greedy IP strategy initially targets layers within the Decisive Phase due to their significant structural redundancy. While the model remains robust during this stage, an abrupt collapse occurs as soon as the pruning process extends into the Silent Phase. We find that these early-to-middle layers serve as the foundational scaffolding for the decision-making process. Removing them, especially after the layers on the Decisive Phase have already been pruned, leaves the model with insufficient depth to achieve the necessary decision transition. Intuitively, the Transition Point is pushed beyond the terminal layer of the pruned network, leaving the Decision Margin permanently negative. This failure to reach the transition state explains why performance collapse is a structural threshold effect: the model is not just losing information, but is rendered architecturally incapable of forming a final decision.

## 2. Related Work

### 2.1. Layer Pruning for LLMs

Layer pruning (Men et al., 2025; Qiao et al., 2025; Wang et al., 2025) is an effective strategy for reducing the computational and memory costs of LLMs. Most existing methods assign importance scores to layers and remove those deemed less critical, using magnitude-based criteria (Kim et al., 2024), loss-based evaluation (Ma et al., 2023), or knowledge-preserving mechanisms such as channel restoration, manifold-guided fusion (e.g., MKA (Liu et al., 2024a)), layer folding (e.g., LaCo (Yang et al., 2024), and (Kou et al., 2026; Wang et al., 2024b)). While these approaches aim at globally optimal pruned models, our iterative pruning framework instead targets *local optimality*, enabling fine-grained analysis of how layer-wise pruning progressively reshapes decision structures in LLMs.

### 2.2. Interpreting Intermediate Representations.

The Logit Lens (Wang, 2025; nostalgebraist, 2020) and Tuned Lens (Belrose et al., 2023) paradigms examine how LLMs incrementally build predictions. While (Wang, 2025) shows that intermediate layers often contain early signals of the final output, (Alain & Bengio, 2016) notes that these readouts can be poorly calibrated. We mitigate this by applying pre-head normalization. Our discovery of a Silent-to-Decisive transition aligns with Mechanistic interpretability studies on Induction Heads (Olsson et al., 2022), where specific circuit formations trigger sudden capability jumps.

## 3. Method

This section formalizes the analytical framework used to investigate the decision-making dynamics of LLMs during layer pruning. We first introduce two decision-level metrics: **Decision Margin (DM)**, which quantifies the model's discriminative confidence, and **Option Frequency (OF)**, which tracks the distributional stability of predictions (Guo et al., 2024). We then present **Iterative Pruning (IP)**, a greedy analytical strategy designed to trace the structural evolution of these metrics as the network depth is progressively reduced.

### 3.1. Decision Margin (DM)

To capture the emergence of decisions across layers, we define the **Decision Margin (DM)**. Unlike traditional metrics that monitor absolute probabilities, DM measures the relative separation between the correct answer and its most potent distractor, providing a direct proxy for the model's discriminative certainty. Formally, for a given layer $l$ and a set of $N$ multiple-choice samples, the Decision Margin is

defined as:

$$\text{DM}(l) = \frac{1}{N} \sum_{i=1}^{N} \left( z_{i,c} - \max_{j \neq c} z_{i,j} \right), \qquad (1)$$

where $z_{i,c}$ denotes the logit of the correct option for sample $i$, and $z_{i,j}$ represents the logit of the $j$-th competing option.

To compute $\text{DM}(l)$ at each layer, we project intermediate hidden states into the vocabulary space using the LM head in a logit-lens framework. To avoid scale distortion caused by distributional shift across layers, we first apply the model's final pre-head normalization layer (e.g., `RMSNorm` or `LayerNorm`) to the intermediate representations prior to projection. This normalization alignment ensures that logits at all depths are produced under a consistent feature scaling regime, making layer-wise DM values directly comparable.

The sign of $\text{DM}(l)$ serves as a functional indicator: a positive value signifies that the correct option has emerged as the leader, whereas a negative value indicates that the model is still dominated by incorrect candidates. We utilize this metric to partition the model into two regimes: the **Silent Phase** (sustained negative DM) and the **Decisive Phase** (sustained positive DM). The layer where the DM crosses the zero-axis is identified as the **Transition Point**.

### 3.2. Option Frequency (OF)

While DM summarizes correctness, it does not reveal the underlying structure of the model's errors. We therefore introduce **Option Frequency (OF)** to characterize the diversity and bias of predicted choices across the candidate space. For a layer $l$ and $M$ possible options, the OF for option $j$ is defined as:

$$\text{OF}(l, j) = \frac{1}{N} \sum_{i=1}^{N} \mathbb{I}(y_i = j), \qquad (2)$$

where $y_i$ is the option index predicted by the model for sample $i$, and $\mathbb{I}(\cdot)$ is the indicator function.

By tracking the evolution of the OF distribution, we can distinguish between a model that is "unsure but balanced" and one that is "trapped in a biased collapse." This allows us to observe how the Silent Phase's internal bias gradually dissolves into the Decisive Phase's structured predictions.

### 3.3. Iterative Pruning (IP)

To achieve a fine-grained observation of structural failure, we propose **Iterative Pruning (IP)**, a greedy, step-wise layer removal strategy. Unlike global pruning methods that optimize for final performance, IP serves as an **analytical probe** to monitor how each individual layer removal shifts the Transition Point.

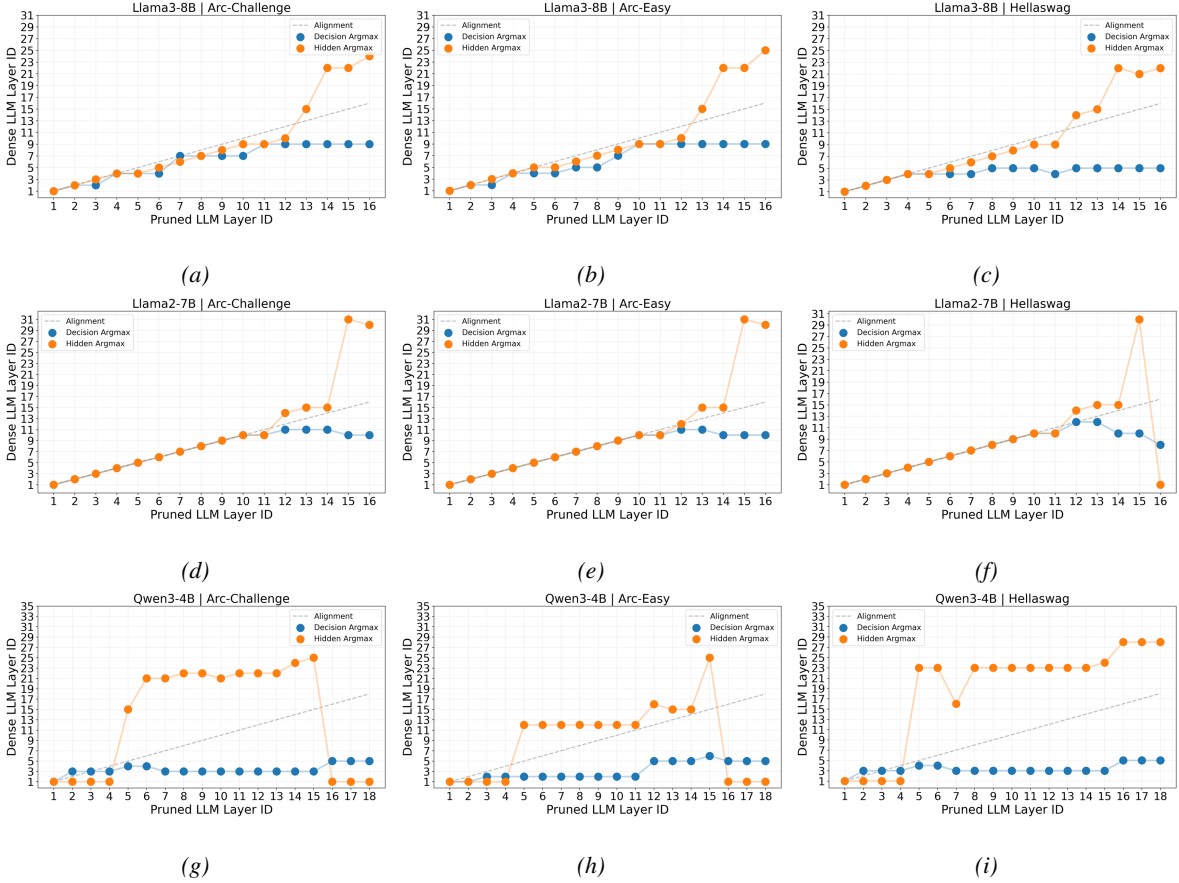

*Figure 3.* Layer-wise CKA alignment between dense and 50%-pruned models across (a-c) Llama3-8B, (d-f) Llama2-7B, and (g-i) Qwen3-4B. The x-axis indexes layers in the pruned model, and the y-axis denotes the dense-model layer with maximum CKA similarity. Orange curves track hidden-state alignment, blue curves track decision-token alignment, and the gray dashed line is the identity correspondence. Under heavy pruning, hidden states can still align with deep dense layers, whereas decision-token representations remain trapped in shallow dense-layer regimes.

**Layer Importance.** At each step, IP evaluates the functional contribution of all remaining layers using the **Block Influence (BI)** (Men et al., 2025) score. To ensure independence, BI is recomputed at every iteration. Given hidden representations at layers $l$ and $l + 1$, the BI score is defined as:

$$\text{BI}_l = \sum_{b=1}^{B} \sum_{s=1}^{S} \Big( 1 - \text{clip}_{[0,1]} \big( \cos(\mathbf{h}_{b,s}^{(l)}, \mathbf{h}_{b,s}^{(l+1)}) \big) \Big), \quad (3)$$

where $\mathbf{h}_{b,s}^{(l)}$ is the representation of the $s$-th token in the $b$-th sequence. A low BI score indicates minimal representational change, identifying the layer as a candidate for removal.

**Greedy Removal.** At iteration $t$, the layer $l^\star$ with the minimum BI score is removed:

$$l^\star = \arg \min_{l \in \mathcal{L}^{(t)}} \text{BI}_l^{(t)}, \quad (4)$$

where $\mathcal{L}^{(t)}$ is the set of layers remaining at step $t$. The model is re-evaluated immediately post-removal to record the shift in DM and OF dynamics.

**SKIP-Prun Extension.** Since greedy strategies can occasionally fall into premature local optima (triggering collapse earlier than a global search would), we introduce **SKIP-Prun**. This mechanism acts as a "restart" trigger: if a specific pruning path leads to immediate and rapid performance deterioration, the trajectory is terminated, and a new IP process is initialized from the best-performing pruned state discovered so far, as in Figure 1 (Bottom). This approach ensures the observation of the **maximum possible pruning trajectory** before the inevitable structural collapse occurs.

## 4. Experiments

### 4.1. Implementation Details

To examine the Silent Phase, Decisive Phase, and the Transition Point across different architectures, we conduct experiments on three representative LLMs: Llama3-8B (Dubey et al., 2024), Llama2-7B (Touvron et al., 2023), and Qwen3-4B (Yang et al., 2025). Our primary evaluation is performed on multiple-choice commonsense reasoning bench-

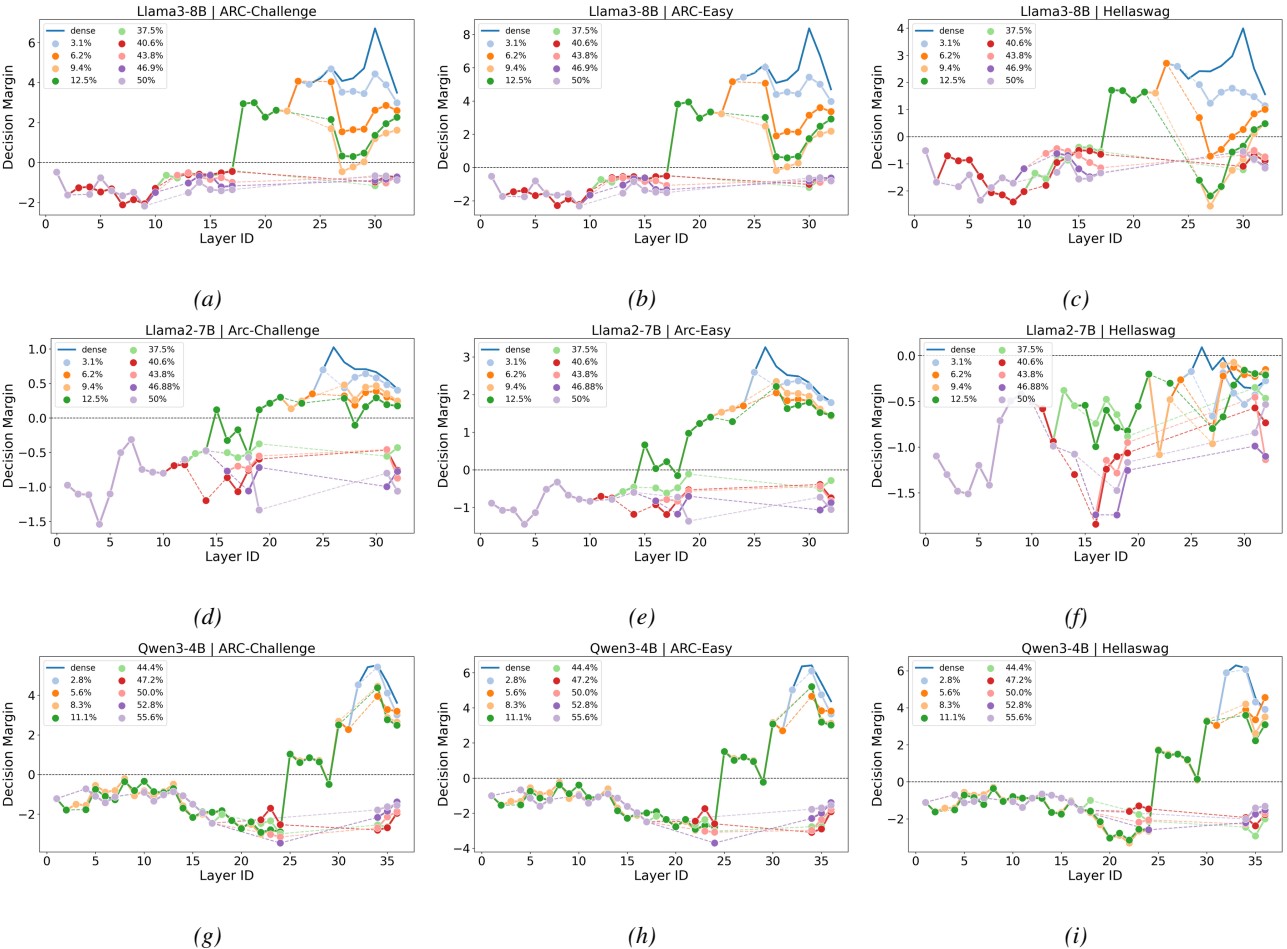

*Figure 4.* Layer-wise Decision Margin (DM) dynamics under progressive pruning across different models and tasks. The "DM jump" characterizes the transition from the Silent Phase ($DM < 0$) to the Decisive Phase ($DM > 0$). Note that performance collapse occurs precisely when pruning encroaches upon the Silent-Phase scaffolding, preventing the formation of a positive margin.

marks, including ARC-Challenge (Clark et al., 2018), ARC-Easy (Clark et al., 2018), and Hellaswag (HellaS) (Zellers et al., 2019). We further validate our findings on additional multiple-choice tasks, including Winogrande (WinoG) (Sakaguchi et al., 2021), PIQA (Bisk et al., 2020), BoolQ (Clark et al., 2019), and MMLU (Hendrycks et al., 2021). More details are provided in the Appendix A.1.

### 4.2. Main Results

#### 4.2.1. THE REPRESENTATION-DECISION DECOUPLING

To investigate the underlying cause of performance collapse, we conduct a comparative analysis of hidden representations versus decision representations using layer-wise CKA alignment. For a pruned model, we identify the functional correspondence by computing the CKA similarity between each pruned layer and all layers of the original dense model. This analysis is performed on two distinct signals: (i) the averaged hidden-layer outputs and (ii) the representations of the decision token.

As illustrated in Figure 3, hidden representations exhibit remarkable resilience to pruning. Despite a 50% pruning rate that triggers total performance collapse, the orange curves show that the remaining layers maintain strong alignment with the deep layers of the dense model. For instance, in Llama3-8B on the ARC-Challenge task, the final layer of the pruned model (Layer 16) aligns most closely with Layer 24 of its dense counterpart. This persistence suggests that the pruned network still preserves high-level semantic features and that performance failure cannot be attributed to the simple disappearance of deep representational structures.

In contrast, decision representations exhibit a profound **representational decoupling**. As shown by the blue curves in Figure 3, the alignment of decision-token representations fails to progress into the deeper regions of the original network, consistently falling below the identity diagonal (the gray dashed line). In the same ARC-Challenge setting, the decision representation at the final pruned layer aligns only with the 9th layer of the dense model.

These results reveal a critical disconnect: while the pruned

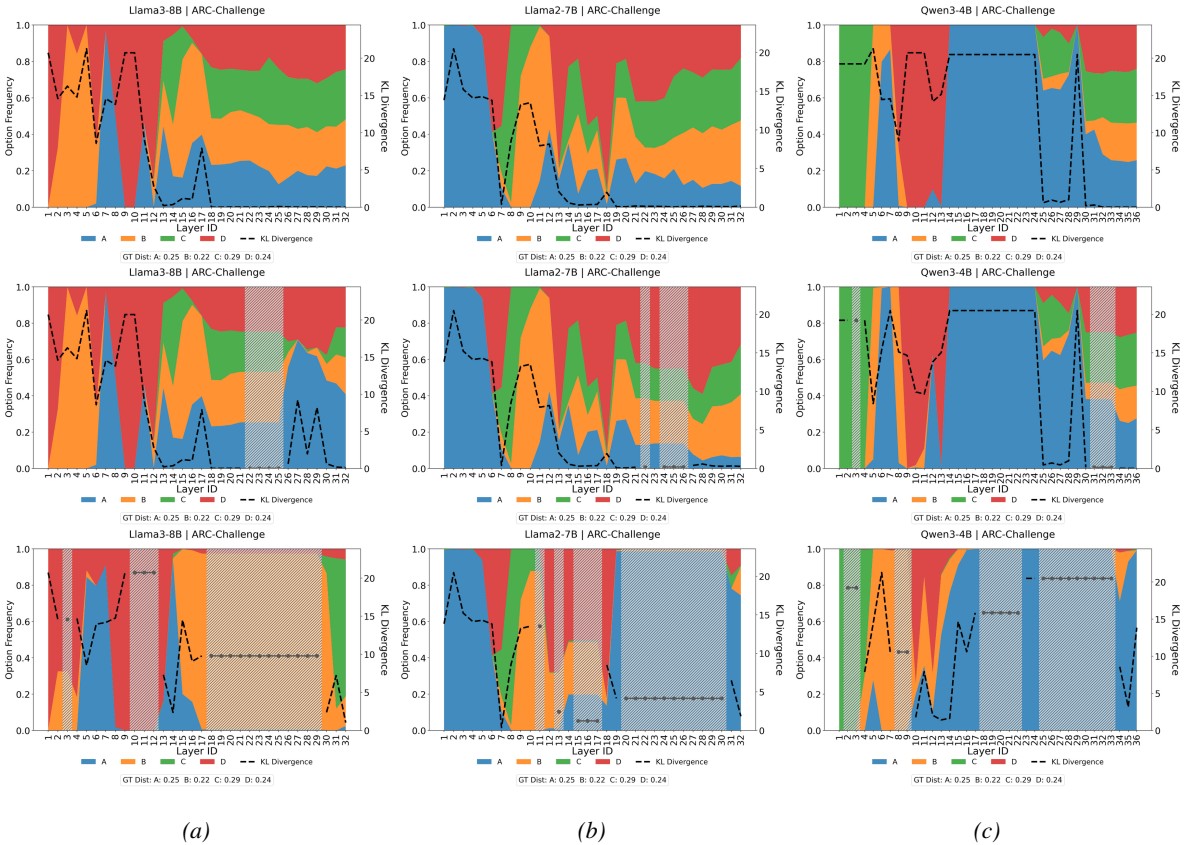

*(a)*             *(b)*             *(c)*

*Figure 5.* Layer-wise KL divergence and option-argmax distributions on ARC-Challenge under progressive pruning. Colored stacked regions A/B/C/D denote the four candidate answer options, and the dashed black curve denotes KL divergence from the ground-truth option distribution shown below each panel. Row 1 shows dense models, Row 2 shows lightly pruned models before collapse, and Row 3 shows heavily pruned models at 50% pruning, where persistent option bias prevents reconstruction of the correct answer distribution.

model retains the "knowledge" stored in deep hidden states, its decision-making logic remains trapped at a stage equivalent to the shallow layers of the original model. This confirms our hypothesis that pruning-induced collapse stems from a structural failure to facilitate the decision transition, rather than a loss of foundational hidden representations. Additional visualizations of decision and hidden representation heatmaps across various tasks are provided in Appendices A.6 and A.7 for further reference.

### 4.2.2. DYNAMICS OF THE DECISION TRANSITION

**The Sharp Decision Transition.** As illustrated in Figure 4, the Decision Margin (DM) in dense LLMs does not accumulate linearly with depth but undergoes a **sharp phase transition**. In the early layers, DM remains consistently negative, indicating that the model cannot yet differentiate the correct answer from distractors. However, once the input reaches a critical depth, *i.e.* the **Transition Point**, the DM abruptly crosses zero and stabilizes at a positive value (e.g., Layer 18 for Llama3-8B, Layer 19 for Llama2-7B, and Layer 30 for Qwen3-4B). This bifurcation naturally partitions the model into a **Silent Phase** and a **Decisive Phase**, suggesting that decision-making is a structurally localized

emergence rather than a gradual refinement.

**Structural Disruption and Collapse.** Layer pruning reveals two distinct regimes of structural sensitivity. When pruning is confined to the **Decisive Phase**, the transition structure remains robust; although the magnitude of DM may decrease, the transition point persists, and the model maintains its correct-option selection. However, as pruning encroaches upon the **Silent Phase**, the model suffers a catastrophic structural failure. In these cases, the DM remains negative throughout the entire network, failing to achieve the transition even at the final output layer. This confirms that collapse arises from the disruption of the foundational decision trajectory, rendering the model permanently unable to form a correct prediction. Notably, in the Hellaswag task, the Decision Margin of Llama2-7B (Figure 4f) remains negative across all layers even in the dense model, which is consistent with its extremely low baseline performance on the dense model, as in Figure 2b.

**The Safe Pruning Margin.** The location of the Transition Point ($T$) dictates the model's inherent robustness to layer removal. Models with a late transition, such as Qwen3-4B ($T = 30$), possess a narrow Decisive Phase ($\sim$16.7% of total depth) and consequently collapse at low pruning ra-

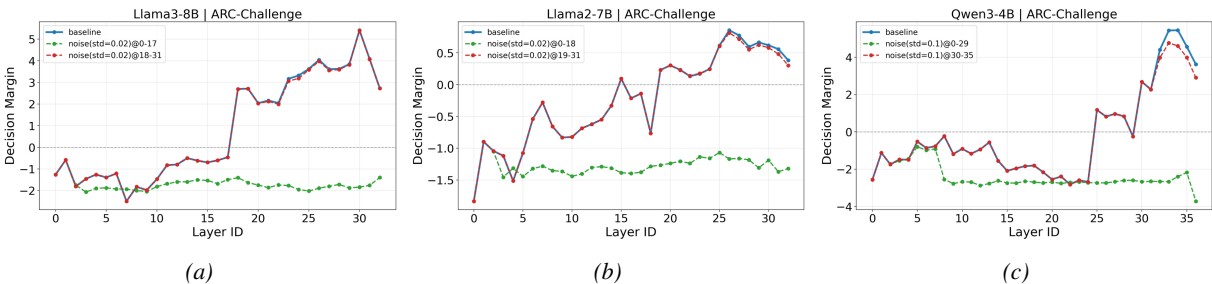

*Figure 6.* Asymmetric noise sensitivity of the Decision Margin across Silent and Decisive Phases. The Silent Phase exhibits extreme fragility to even minor perturbations (variance 0.02/0.5), whereas the Decisive Phase demonstrates structural robustness, maintaining its discriminative margin under significantly larger noise magnitudes.

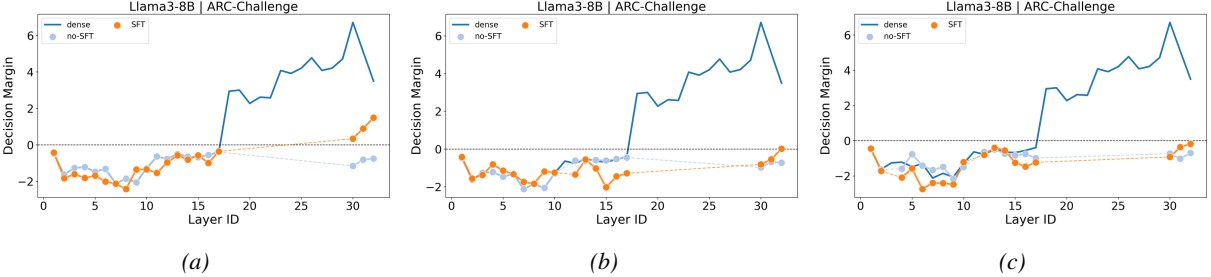

*Figure 7.* Impact of Supervised Fine-Tuning (SFT) on the layer-wise DM of pruned Llama3-8B models. While SFT enhances discriminative confidence in moderate pruning regimes (37.5%), it fails to reconstruct the necessary structural depth required to trigger the decision transition in heavily pruned models (40.6% and 43.8%).

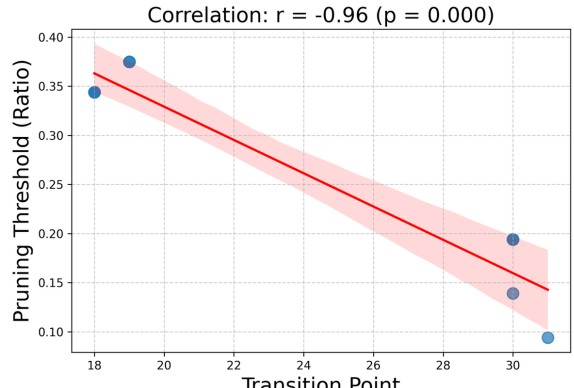

*Figure 8.* Quantitative correlation between the Decision Margin (DM) transition point and model pruning robustness. The x-axis represents the transition layer index, and the y-axis indicates the critical pruning threshold. A negative correlation ($r = -0.96, p < 0.001$) between these two metrics confirms that a later transition point correlates with higher susceptibility to performance collapse.

tios. In contrast, Llama2-7B and Llama3-8B exhibit earlier transitions ($T \approx 18$-$19$), providing a larger structural buffer ($\sim 40\%$ of depth) that allows them to tolerate significantly higher pruning rates (Figure 2).

To quantitatively evaluate the predictive power of the DM transition point, we perform a Pearson correlation analysis across all model-task pairs. As in Figure 8, the results show a significant negative correlation ($r = [-0.96], p < [0.001]$) between the transition layer index and the critical pruning threshold. This indicates that the depth of the Decisive Phase serves as the strong preliminary evidence for a model's struc-

tural redundancy: an earlier transition into the "Decisive Phase" grants greater pruning robustness.

**Protocol-level controls.** To test whether the transition-collapse pattern is an artifact of our BI-based iterative trajectory, we compare IP with one-shot BI pruning (ShortGPT-style), manifold-based pruning (MKA), and random pruning in Table 1. The same qualitative pattern holds across non-random protocols: accuracy remains relatively stable while the DM transition exists, and collapses when the transition disappears. Random pruning destroys the transition at much lower pruning ratios, indicating that the transition structure is not merely a byproduct of the IP trajectory. These results support the transition disappearance as a dominant structural signal of pruning-induced collapse, while not ruling out other contributing mechanisms.

### 4.2.3. OPTION DYNAMICS AND DISTRIBUTIONAL RECONSTRUCTION

**Stability in the Decisive Phase.** Using the **Option Frequency (OF)** metric, we analyze how the distribution of predicted options aligns with the ground-truth label distribution (Kou et al., 2024; 2025a;b) via KL divergence. In the Decisive Phase of dense models, the predicted distribution is remarkably stable and closely mirrors the true label distribution ($KL \approx 0$). As shown in Figure 5 (Line 1), once the model crosses the transition point, it recovers a statistically uniform and accurate representation of the task's option space.

**Latent Reorganization in the Silent Phase.** While "silent"

*Table 1.* Transition-collapse coupling under alternative pruning protocols on Llama3-8B. NT denotes the first pruning ratio where no DM transition is observed, CP denotes the first collapse ratio, and Acc($C-10$), Acc($C$), and Acc($C+10$) denote accuracies before, at, and after collapse. Across non-random pruning criteria, collapse coincides with transition disappearance; random pruning removes the transition at much lower ratios.

| Task | Method | NT (%) | CP (%) | Acc($C-10$) | Acc($C$) | Acc($C+10$) |
|---|---|---|---|---|---|---|
| ARC-C | IP | 40 | 40 | 0.80 | 0.28 | 0.25 |
| | ShortGPT | 40 | 40 | 0.47 | 0.23 | 0.21 |
| | MKA | 50 | 50 | 0.80 | 0.27 | – |
| | Random | 10 | 10 | 0.82 | 0.21 | 0.20 |
| HellaS | IP | 40 | 40 | 0.64 | 0.28 | 0.26 |
| | ShortGPT | 30 | 30 | 0.70 | 0.27 | 0.27 |
| | MKA | 50 | 50 | 0.78 | 0.28 | – |
| | Random | 10 | 10 | 0.76 | 0.20 | 0.22 |

in terms of accuracy, the pre-transition layers exhibit a crucial two-stage internal reorganization. In the initial layers (e.g., Layers 1-13 of Llama3-8B), we observe a **distributional collapse** where predictions are dominated by a single option across all samples, reflecting a strong internal bias. Subsequently, a **diversification stage** emerges (Layers 13-18), where the model begins to disentangle option representations and distribute probability mass across the candidate space. This implies that the Silent Phase is not functionally void but acts as a necessary pipeline for **structural disentanglement**, preparing the representation for the final decision transition.

**Failure of Global Reconstruction.** Under pruning, the failure to maintain a correct output is essentially a failure of **global distributional reconstruction**. When pruning is limited, the model undergoes transient perturbations in remaining last layers but successfully reconstructs the answer distribution by the final layer (Figure 5 (Line 2)). However, once the Silent Phase is pruned after the Decisive Phase is excessively thinned, the model lacks the functional depth required to overcome its initial bias. The resulting output remains trapped in a biased state, characterized by a sharp spike in KL divergence and total performance collapse. This demonstrates that pruning-induced failure is not a local artifact but a systemic inability to reconstruct a valid decision space. Discrete radar plots of these dynamics are available in Appendix A.9 for better visualizations.

### 4.3. Ablation Studies

#### 4.3.1. PHASE-DEPENDENT SENSITIVITY TO STOCHASTIC PERTURBATIONS

To evaluate the structural stability of the identified phases, we introduce random noise into the layer-wise hidden states and monitor the impact on the Decision Margin (DM). We normalize the injected noise by the layer-wise standard deviation, i.e., $\mathbf{h}'_l = \mathbf{h}_l + \epsilon \cdot \sigma(\mathbf{h}_l) \cdot \mathcal{N}(0, \mathbf{I})$ for fair com-

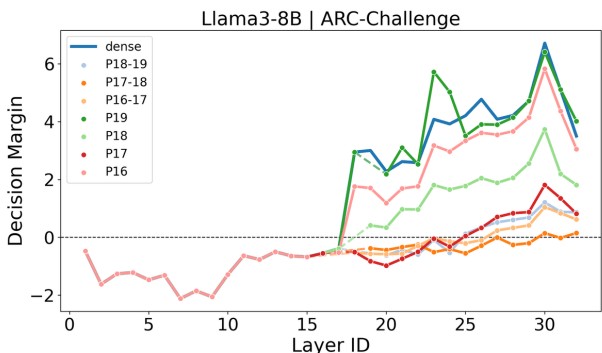

*Figure 9.* Structural ablation analysis around the Transition Point (Layer 18) for Llama3-8B. Removing layers proximal to the transition boundary (Layers 17-18) causes a catastrophic drop in Decision Margin, whereas removing layers deeper within the Decisive Phase (e.g., Layer 19) results in negligible degradation.

parison across layers with different activation magnitudes. As shown in Figure 6, we observe a striking **asymmetric vulnerability** between the two regimes. The Silent Phase exhibits extreme fragility: even low-magnitude noise ($\sigma^2 = 0.02$) significantly degrades the DM, indicating that representations in this region are in a highly volatile state of formation. Conversely, the Decisive Phase demonstrates remarkable resilience, maintaining a stable DM even under substantially higher noise levels (e.g., $\sigma^2 = 0.1$ in Qwen3-4B). This disparity confirms that the Silent Phase is the model's structural Achilles' heel, whereas the Decisive Phase forms a robust decision regime. Such observations explain why structural interventions like pruning trigger irreversible collapse primarily when they encroach upon the Silent Phase. The same observation are also observed in the case of the ARC-Easy and Hellaswag tasks, as in Appendix A.10.

*Table 2.* Zero-shot accuracy (%) of Llama3-8B under targeted single-layer and two-layer ablations. The most severe degradation occurs when the "Silent-to-Decisive" bottleneck (Layers 17 or 17-18) is disrupted, while ablating Decisive-Phase layers (Layer 19) maintains high performance, confirming the phase-dependent criticality.

| Task | 16 | 17 | 18 | 19 | 16,17 | 17,18 | 18,19 |
|------|----|----|----|----|-------|-------|-------|
| ARC-C | 82.76 | 72.27 | 81.66 | **82.68** | 63.48 | 55.03 | 65.70 |
| ARC-E | 92.76 | 84.67 | 91.24 | **93.09** | 69.64 | 68.04 | 80.42 |
| HellaS | 68.48 | 47.39 | 65.04 | **71.92** | 50.10 | 37.62 | 50.95 |
| PIQA | 80.30 | 80.58 | **80.90** | 79.92 | 78.73 | 77.09 | 79.49 |
| MMLU | 64.40 | 56.98 | 61.41 | **64.71** | 48.86 | 41.42 | 54.48 |
| WinoG | 57.22 | 58.09 | 56.83 | **58.09** | 53.04 | 57.46 | 57.22 |
| BoolQ | 77.89 | **81.22** | 78.62 | 80.95 | 76.79 | 77.98 | 77.83 |
| Avg | 74.83 | 68.74 | 73.67 | **75.91** | 62.95 | 59.24 | 66.58 |

*Table 3.* A simple Transition-Aware Pruning (TAP) control on Llama3-8B. TAP applies the same BI ranking while protecting a narrow band around the dense transition index $[T - 1, T, T + 1]$. The results show that transition protection can improve robustness in several medium/high-pruning regimes, although it is not uniformly better across all tasks and ratios.

| Task | Ratio | BI Acc | TAP Acc | $\Delta$ |
|------|-------|--------|---------|----------|
| ARC-C | 0% | 0.81 | 0.81 | 0.00 |
| | 10% | 0.77 | 0.77 | 0.00 |
| | 20% | 0.68 | 0.76 | +0.08 |
| | 40% | 0.28 | 0.41 | +0.13 |
| HellaS | 0% | 0.76 | 0.76 | 0.00 |
| | 20% | 0.45 | 0.58 | +0.13 |
| | 40% | 0.29 | 0.21 | -0.08 |

### 4.3.2. THE TRANSITION POINT AS A STRUCTURAL BOTTLENECK

We further investigate the criticality of the transition region by selectively ablating layers around the Transition Point (Layer 18 for Llama3-8B). As illustrated in Figure 9, removing layers produces highly asymmetric consequences. Ablating the **pre-transition pair** (Layers 17-18) induces the most severe collapse, driving the DM below zero. At a single-layer granularity, removing Layer 17 (late Silent Phase) causes a much sharper performance decline than removing Layer 19 (early Decisive Phase). The same results on the ARC-Easy and Hellaswag tasks are shown in A.11.

These results, corroborated by the zero-shot accuracy in Table 2, pinpoint the transition region as the **bottleneck** for decision formation. Disrupting this juncture prevents the model from achieving the necessary "jump" in discriminative confidence. While perturbations in the Decisive Phase merely weaken the final decision's magnitude, interfering with the Transition Point effectively destroys the decision-making mechanism itself.

The previous analysis suggests a concrete structural constraint: preserve the transition-supporting band while pruning more redundant layers elsewhere. To verify this , we propose a method called Transition-Aware Pruning (TAP) to identify the dense model's transition index $T$, and protect $[T - 1, T, T + 1]$. We then apply the same BI ranking to the remaining layers. Table 3 shows that this simple constraint improves robustness on ARC-C at 20% and 40% pruning and on HellaSwag at 20%. At 40%, both pruning methods exhibit performance collapse on HellaSwag, while TAP mitigates the performance degradation of ARC-C (0.41 vs. 0.28). Therefore, TAP can be viewed as evidence that the transition structure can provide actionable pruning constraints.

### 4.3.3. BOUNDED RECOVERY VIA SUPERVISED FINE-TUNING

Finally, we examine whether the performance collapse in heavily pruned models (37.5%-43.8% pruning ratios) can be mitigated through Supervised Fine-Tuning (SFT) (Liu et al., 2024b). While SFT consistently improves the DM across all configurations (Figure 7), the recovery is strictly **bounded by the remaining structural depth**. At moderate pruning (37.5%), SFT can partially restore the decision transition. However, as pruning depth increases (40.6% and beyond), the improvement plateaus, and the model remains unable to cross the transition threshold. This suggests that SFT can optimize existing paths to some extent, but cannot reconstruct the **essential structural scaffolding** lost during excessive pruning of the Silent Phase.

## 5. Conclusion

In this work, we provide a decision-centric interpretation of the performance collapse observed in layer-pruned Large Language Models (LLMs). By operationalizing two decision-level metrics, **Decision Margin (DM)** and **Option Frequency (OF)**, we reveal a stage-structured decision hierarchy consisting of a foundational **Silent Phase** and a **Decisive Phase**. Our analysis, facilitated by **Iterative Pruning (IP)** as an analytical probe, shows that collapse is not adequately explained by gradual hidden-representation loss alone. Instead, in our evaluated settings, a dominant failure mode is the removal of critical decision scaffolding in the Silent Phase, which prevents the model from crossing the **Transition Point** to form a discriminative decision. These findings provide structural guidance for transition-aware pruning while leaving broader pool-free open-ended generation and other compression mechanisms as future work.

## Acknowledgments

This research was supported by the Jiangsu Science Foundation (BK20243012, BG2024036); the National Science Foundation of China (62125602, U24A20324, 92464301); the New Cornerstone Science Foundation through the XPLORER PRIZE; the Fundamental Research Funds for the Central Universities (2242025K30024); the National Natural Science Foundation of China (62576091); the Southeast University Big Data Computing Center and Southeast University Kunpeng & AscendCenter of Cultivation.

## Impact Statement

This research enhances the structural understanding and efficiency of LLMs through a systematic analysis of decision formation and layer-pruning dynamics. By identifying the critical layers necessary for stable decision-making, our work contributes to more reliable model compression techniques, potentially reducing the computational costs and environmental footprint associated with deploying large-scale AI. All experiments were conducted using publicly available datasets and models, involving no sensitive personal information or human subjects. As a foundational study in machine learning, the downstream societal effects are indirect, and no specific ethical risks or misuses are identified beyond the general challenges of AI safety and responsible deployment.

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

# A. Appendix

We organize the appendix as follows.

- Appendix A.1: Dataset specifications and experimental hyperparameters.

- Appendix A.3: Extra related works (Learngene) for identifying the critical layers.

- Appendix A.4: Pseudo-code for Iterative Pruning (IP) with SKIP-Prun.

- Appendix A.5: Records of specific layer IDs removed across different models.

- Appendix A.6: CKA analysis of hidden representations (ARC tasks).

- Appendix A.7: Heatmap similarity of decision-token representations.

- Appendix A.8: Detailed DM dynamics under full pruning trajectories.

- Appendix A.9: Layer-wise option frequency and radar plot visualizations.

- Appendix A.10: Phase-dependent noise sensitivity evaluations.

- Appendix A.11: Detailed ablation studies around the Transition Point.

- Appendix A.12: Performance comparison of IP against state-of-the-art pruning baselines.

- Appendix A.13: Discussion of limitations and future research directions.

## A.1. Experimental Details

### A.1.1. DATASET SPECIFICATIONS

**ARC (Challenge & Easy):** The AI2 Reasoning Challenge consists of grade-school science questions. The *Challenge* set is filtered to include only questions that require complex reasoning beyond simple retrieval, making it the primary benchmark for our decision-transition analysis.

**Hellaswag:** A commonsense reasoning benchmark focused on scenario completion. It uses adversarial filtering to ensure models rely on contextual coherence rather than surface-level patterns.

**WinoGrande:** A coreference resolution task requiring background knowledge to resolve ambiguous pronouns in sentence-completion formats.

**BoolQ:** A binary (Yes/No) question-answering task based on real-world queries and supporting passages, evaluating grounded comprehension.

**MMLU:** A multi-task benchmark covering 57 subjects across STEM, social sciences, and humanities, testing broad knowledge integration.

**PIQA:** Focuses on physical commonsense reasoning, requiring an understanding of object affordances and physical plausibility in everyday tasks.

### A.1.2. HYPERPARAMETERS AND SAMPLING

**Evaluation Sampling:** For BI score computation and CKA alignment, we use 256 and 500 randomly sampled instances from ARC-Challenge, respectively. This ensures a stable statistical signal for layer-wise measurements while maintaining computational efficiency.

**Supervised Fine-Tuning (SFT):** SFT is performed on 4 GPUs with an effective batch size of 8 and a max sequence length of 512. We train for 3 epochs using AdamW. The learning rate is adaptively selected from $[1 \times 10^{-5}, 5 \times 10^{-5}]$ to prevent spurious optimization collapse, ensuring the model's recovery reflects its structural capacity rather than suboptimal training. No test-set fine-tuning or evaluation-set leakage is used in our SFT comparisons. The SFT/evaluation splits are BoolQ train/validation, ARC-Easy train/test, ARC-Challenge train/test, PIQA train/validation, HellaSwag train/validation, and WinoGrande xl_train/xl_validation. For MMLU, SFT uses the train split when available and otherwise the validation split, while evaluation uses the test split.

*Table 4.* Post-hoc open-generation diagnostic on GSM8K. We construct an answer pool containing the gold answer and negative candidates only for layer-wise analysis, not as a pruning protocol. EM denotes exact match, and T idx denotes the layer where the gold-answer margin becomes positive.

| Model | Ratio | EM | Final DM | T idx |
|---|---|---|---|---|
| Llama3-8B | Dense | 0.66 | 0.2365 | 23 |
| | 10% | 0.37 | 0.0086 | 23 |
| | 20% | 0.00 | -0.1674 | – |
| | 30% | 0.00 | -0.2166 | – |
| | 40% | 0.00 | -0.3064 | – |
| | 50% | 0.00 | -0.4510 | – |
| Qwen3-4B | Dense | 0.74 | 0.7486 | 29 |
| | 10% | 0.68 | 0.4645 | 29 |
| | 20% | 0.01 | -0.0862 | – |
| | 30% | 0.00 | -0.4285 | – |
| | 40% | 0.00 | -0.2366 | – |
| | 50% | 0.00 | -0.3660 | – |

## A.2. Open-generation Diagnostic on GSM8K

Table 4 provides a post-hoc diagnostic beyond standard multiple-choice tasks. As pruning increases, EM drops sharply and the final-layer DM becomes negative when the transition disappears. This suggests that transition-like behavior can be observed beyond visible-option MCQ settings when a reasonable candidate-answer surrogate is available. However, this answer-pool construction is only an analysis tool and should not be interpreted as a practical pruning requirement for fully open-ended generation; pool-free decision diagnostics remain future work.

## A.3. Extended Related Works

Learngene (Wang et al., 2022; Shi et al., 2024) aim to extract compact and reusable subnetworks (*learngenes*) from large ancestor models to initialize smaller descendant models (Shi et al., 2026b) with minimal fine-tuning, and can be viewed as different strategies for identifying structurally important layers in deep models. Vanilla Learngene (Van-LG) (Wang et al., 2022) selects high-level layers based on gradient importance, while Auto-Learngene (Wang et al., 2023) learns layer selection through meta-networks and pseudo DesNets. Learngene Pool (Shi et al., 2024) aggregates candidate layers via multi-student distillation, and (Xia et al., 2024a; Wang et al., 2024a; Xia et al., 2024b) further extend learngenes through scalable expansion or modular stage-based structures. These works share a common goal of locating key layers for model compression and transfer, but provide limited analysis of the decision-level structures in LLMs.

## A.4. Algorithm for the Analysis Process

To ensure reproducibility, we provide the detailed pseudo-code for the **Iterative Pruning (IP)** algorithm with **SKIP-Prun restart** in Algorithm 1. This algorithm integrates the computation of **Decision Margin (DM)** and **Option Frequency (OF)** to monitor the model's structural evolution (Guo et al., 2025) in real-time during the pruning process.

## A.5. Removed Layer IDs Across Pruning Ratios

This section documents the exact indices of layers removed at various pruning stages for Llama3-8B (Table 5), Llama2-7B (Table 6), and Qwen3-4B (Table 7) on the ARC-Challenge task.

Consistent with our SKIP-Prun strategy, the pruning paths are dynamically adjusted to bypass local optima that trigger premature collapse. For instance, the pruning configuration for Llama3-8B at a 25% ratio may not be a simple subset of the 21.875% configuration; such divergence indicates that a restart was triggered to find a more robust structural trajectory. These tables provide a complete audit trail of the layer-level modifications used in our study.

### A.6. Hidden Representation Similarity (CKA)

We evaluate the CKA similarity of hidden representations between pruned (50% ratio) and dense models on ARC tasks (Figure 10). Our results consistently show that even at extreme pruning ratios that cause total performance collapse, the remaining layers in the pruned LLM maintain high CKA alignment with the *deep* layers of the original dense model. This confirms that hidden semantic features are largely preserved, and the performance cliff is not driven by the disappearance of deep-layer hidden states.

### A.7. Decision Representation Similarity Heatmaps

As illustrated in Figure 11, the similarity heatmaps for decision-token representations reveal a distinct two-phase hierarchy. While shallow layers maintain high similarity across pruned and dense models (bright regions), the deeper layers exhibit a sharp disconnect (dark regions). This suggests that 50% pruning effectively **truncates the decision hierarchy**, leaving the model with only the foundational, shallow-layer decision logic.

A critical observation occurs in Llama2-7B on the Hellaswag task (Figure 11 (b)). Despite maintaining high similarity in deep-layer decision tokens, the model's accuracy remains at random levels. This highlights a fundamental limitation of similarity-based metrics: **high representation similarity does not necessarily imply effective decision discrimination.** In cases of low confidence or persistent indecision, both models may share a similar, yet functionally useless, output distribution. This underscores the necessity of the **Decision Margin (DM)** as a complementary metric to distinguish between "preserved structure" and "preserved capability."

### A.8. Decision Margin Dynamics Under Full Pruning

We analyze the $DM$ evolution across all pruning ratios up to 60% (Figure 12). For Llama3-8B and Qwen3-4B, the final-layer $DM$ decreases monotonically until it hits a critical threshold, after which it stays permanently negative, signaling the structural inability to form a correct decision. In the case of Llama2-7B on Hellaswag, the $DM$ is negative from the outset, reflecting its initial lack of task-specific discriminative power, which further degrades under pruning.

Table 8 evaluates whether the transition point is sensitive to sampled evaluation subsets, prompt templates (Wang et al., 2024c), or option-order perturbations. Across all three settings, the transition exists with rate 1 before collapse and is consistently located at layer 18, whereas it disappears with rate 0 in the collapsed 40%-50% pruning regimes. The final-layer DM is also consistently positive before collapse and negative after collapse, indicating that the transition estimate is not a fragile artifact of a single subset, prompt, or option order.

Table 9 shows that the DM transition is not fixed at the same layer across all domains. Tasks within a related domain can have nearby transition layers, but different domains may shift the transition or fail to produce a positive transition under the tested model. This motivates treating the transition point as an empirical structural property of a model-task pair, not as a universal architectural constant.

### A.9. Extended Option Distribution Analysis

We provide a dual-perspective analysis of option distributions: (i) layer-wise KL divergence/argmax distributions (Figures 13 and 14) and (ii) discrete radar plots (Figures 15 to 17).

For dense models, the transition from the biased Silent Phase to the stable Decisive Phase is characterized by the KL divergence converging to zero. Pruning limited to the Decisive Phase allows the model to eventually reconstruct a valid distribution. However, encroaching upon the Silent Phase prevents this reconstruction, trapping the model in a state of high bias. The radar plots further visualize this by showing how the "probability mass" fails to expand into the correct option space under excessive pruning.

Because OF could be confounded by answer-position imbalance or prompt-specific option bias, we summarize OF by entropy under subset, prompt, and option-order perturbations. Table 10 shows a stable pattern: the mid-Silent region remains more concentrated, the late-Decisive region is high-entropy before collapse, and collapsed models lose this high-entropy Decisive behavior. Since entropy is invariant to option renaming, this control suggests that OF reflects phase-level decision dynamics rather than only superficial option labels.

**Lens dependence.** Table 11 compares raw LogitLens, normalization-aligned lens, and affine lens readouts. All three lenses identify the same transition layer and produce similar final accuracies, with high cross-lens top-1 agreement. Thus, while the decoding lens can affect the strength of option preferences, it does not explain away the phase behavior or transition location.

### A.10. Phase-Dependent Noise Sensitivity

Extended noise sensitivity tests on ARC-Easy and Hellaswag (Figure 18) corroborate our main finding: the **Silent Phase is significantly more fragile** than the Decisive Phase. Small stochastic perturbations in the Silent Phase lead to disproportionate collapses in $DM$, confirming its role as a delicate structural scaffold that is sensitive to any form of modification.

### A.11. Transition Point Criticality

Ablation studies around the Transition Point on ARC-Easy and Hellaswag (Figure 19) show that removing the Transition Point or its immediate Silent-Phase predecessors (e.g., Layer 17 in Llama3-8B) is catastrophic. This reinforces the idea that the "jump" in decision confidence is concentrated at this structural bottleneck.

To further complement the single-layer and two-layer ablations, we remove contiguous three-layer windows across the Silent, transition, and Decisive regions. Table 12 shows that deleting Silent or transition-supporting windows leads to substantially lower accuracy, whereas deleting Decisive-phase windows preserves much higher performance. This broader window-level control supports our interpretation that the transition-supporting scaffold is a dominant vulnerability under layer removal.

### A.12. Baseline Pruning Comparisons

**SFT baseline comparison.** We compare Iterative Pruning (IP) with state-of-the-art baselines including ShortGPT, SLEB, and MKA on Llama3-8B (Table 13). IP consistently yields superior post-SFT performance across all tasks. We attribute this to the **adiabatic nature** of incremental, single-layer removal. By allowing the model to adapt to minimal structural shifts at each step, IP avoids the massive representational shocks inherent in global or multi-layer pruning methods, thereby preserving more of the learned decision scaffolding.

**Zero-shot baseline comparison.** Table 14 reports a direct zero-shot comparison between ShortGPT-style one-shot pruning and IT-Prun. The results are intentionally not presented as universal superiority of IT-Prun. Instead, they show that pruning behavior depends on the model, ratio, and task: ShortGPT performs better at 10% and 20% on Llama3-8B, while IT-Prun retains substantially higher accuracy at 30% and on Qwen3-4B at 10% and 20%. This supports our use of IT-Prun as an analytical path for tracing decision dynamics, rather than as the sole claim of algorithmic dominance.

### A.13. Limitations and Future Work

While this work identifies a clear decision transition in layer-pruned LLMs, it is primarily focused on **multiple-choice reasoning tasks** where decision outcomes are discrete and well-defined. Future work could extend this framework to **open-ended generation**, where the "Transition Point" may manifest as the emergence of semantic coherence or syntactic stability. Additionally, while we focus on **layer-wise depth pruning**, exploring how width-wise sparsity or quantization affects the Decision Margin could provide a more holistic view of how model compression disrupts the decision-making scaffolding. Investigating whether the Transition Point can be "shifted" or "compressed" during the pre-training phase remains an open and promising research direction. Finally, width pruning shows distributed degradation and needs separate mechanism study

---

**Algorithm 1** Iterative Pruning (IP) with SKIP-Prun Restart

---

**Require:** Dense LLM $M_{\text{dense}}$ with layers $\mathcal{L}_{\text{dense}}$; Calibration set $\mathcal{D}_{\text{calib}}$; Target depth $L_{\text{target}}$; Collapse threshold $\tau$.
**Ensure:** Pruned model $M$ of depth $L_{\text{target}}$.
1: $M \leftarrow M_{\text{dense}}, \mathcal{L} \leftarrow \mathcal{L}_{\text{dense}}, L_{\text{anchor}} \leftarrow |\mathcal{L}_{\text{dense}}|$
2: $\text{Acc}_{\text{prev}} \leftarrow \text{Acc}(M, \mathcal{D}_{\text{calib}})$
3: **while** $|\mathcal{L}| > L_{\text{target}}$ **do**
4:     {Compute layer-wise importance}
5:     **for all** $l \in \mathcal{L}$ **do**
6:         $\text{BI}_l \leftarrow$ Compute Block Influence using Eq.(4)
7:     **end for**
8:     {Greedy removal step}
9:     $l^\star \leftarrow \arg\min_{l \in \mathcal{L}} \text{BI}_l$
10:    $M' \leftarrow$ Prune $l^\star$ from $M$; $\mathcal{L}' \leftarrow \mathcal{L} \setminus \{l^\star\}$
11:    $\text{Acc}_{\text{cur}} \leftarrow \text{Acc}(M', \mathcal{D}_{\text{calib}})$
12:    {SKIP-Prun Check}
13:    **if** $(\text{Acc}_{\text{prev}} - \text{Acc}_{\text{cur}}) > \tau$ **then**
14:       $L_{\text{anchor}} \leftarrow |\mathcal{L}'|$
15:       **Restart:** $M \leftarrow$ Prune $(L_{\text{dense}} - L_{\text{anchor}})$ layers globally from $M_{\text{dense}}$
16:       $\mathcal{L} \leftarrow$ Update layer set to current state
17:       $\text{Acc}_{\text{prev}} \leftarrow \text{Acc}(M, \mathcal{D}_{\text{calib}})$
18:    **else**
19:       $M \leftarrow M', \mathcal{L} \leftarrow \mathcal{L}', \text{Acc}_{\text{prev}} \leftarrow \text{Acc}_{\text{cur}}$
20:    **end if**
21:    Log metrics $\{DM, OF\}$ for $M$
22: **end while**
23: Return $M$

---

*Table 5.* Detailed record of 0-based layer indices removed from Llama3-8B during the Iterative Pruning (IP) process on ARC-Challenge. The pruning ratio is defined relative to the original 32-layer architecture. These trajectories illustrate the specific structural modifications analyzed in our study.

| Remain | #Removed | Pruning Ratio | Removed Layer IDs (0-based) |
|--------|----------|---------------|----------------------------|
| 31 | 1 | 3.125% | [24] |
| 30 | 2 | 6.250% | [24, 23] |
| 29 | 3 | 9.375% | [24, 23, 22] |
| 28 | 4 | 12.500% | [24, 23, 22, 21] |
| 27 | 5 | 15.625% | [24, 23, 22, 21, 19] |
| 26 | 6 | 18.750% | [24, 23, 22, 21, 19, 20] |
| 25 | 7 | 21.875% | [24, 23, 22, 21, 19, 20, 18] |
| 24 | 8 | 25.000% | [24, 23, 25, 26, 27, 28, 22, 21] |
| 23 | 9 | 28.125% | [24, 23, 25, 26, 27, 28, 22, 21, 19] |
| 22 | 10 | 31.250% | [24, 23, 25, 26, 27, 28, 22, 21, 19, 20] |
| 21 | 11 | 34.375% | [24, 23, 25, 26, 27, 28, 22, 21, 19, 20, 18] |
| 20 | 12 | 37.500% | [24, 23, 25, 26, 27, 28, 22, 21, 19, 20, 18, 17] |
| 19 | 13 | 40.625% | [24, 23, 25, 26, 27, 28, 22, 21, 19, 20, 18, 17, 10] |
| 18 | 14 | 43.750% | [24, 23, 25, 26, 27, 28, 22, 21, 19, 20, 18, 17, 10, 2] |
| 17 | 15 | 46.875% | [24, 23, 25, 26, 27, 28, 22, 21, 19, 20, 18, 17, 10, 2, 11] |
| 16 | 16 | 50.000% | [24, 23, 25, 26, 27, 28, 22, 21, 19, 20, 18, 17, 10, 2, 11, 9] |

*Table 6.* Detailed record of 0-based layer indices removed from Llama2-7B during the Iterative Pruning (IP) process on ARC-Challenge. The pruning ratio is defined relative to the original 32-layer architecture. These trajectories illustrate the specific structural modifications analyzed in our study.

| Remain | #Removed | Pruning Ratio | Removed Layer IDs (0-based) |
|--------|----------|---------------|------------------------------|
| 31 | 1 | 3.125% | [25] |
| 30 | 2 | 6.250% | [25, 24] |
| 29 | 3 | 9.375% | [25, 24, 23] |
| 28 | 4 | 12.500% | [25, 24, 23, 21] |
| 27 | 5 | 15.625% | [25, 24, 23, 21, 20] |
| 26 | 6 | 18.750% | [25, 24, 23, 21, 20, 26] |
| 25 | 7 | 21.875% | [25, 24, 23, 21, 20, 26, 19] |
| 24 | 8 | 25.000% | [25, 24, 23, 21, 20, 26, 19, 22] |
| 23 | 9 | 28.125% | [25, 24, 23, 26, 21, 20, 27, 28, 29] |
| 22 | 10 | 31.250% | [25, 24, 23, 26, 21, 20, 27, 28, 29, 19] |
| 21 | 11 | 34.375% | [25, 24, 23, 26, 21, 20, 27, 28, 29, 19, 22] |
| 20 | 12 | 37.500% | [25, 24, 23, 26, 21, 20, 27, 28, 29, 19, 22, 14] |
| 19 | 13 | 40.625% | [25, 24, 23, 26, 21, 20, 27, 28, 29, 19, 22, 14, 12] |
| 18 | 14 | 43.750% | [25, 24, 23, 26, 21, 20, 27, 28, 29, 19, 22, 14, 12, 10] |
| 17 | 15 | 46.875% | [25, 24, 23, 26, 21, 20, 27, 28, 29, 19, 22, 14, 12, 10, 16] |
| 16 | 16 | 50.000% | [25, 24, 23, 26, 21, 20, 27, 28, 29, 19, 22, 14, 12, 10, 16, 15] |

*Table 7.* Detailed record of 0-based layer indices removed from Qwen3-4B during the Iterative Pruning (IP) process on ARC-Challenge. The pruning ratio is defined relative to the original 36-layer architecture. These trajectories illustrate the specific structural modifications analyzed in our study.

| Remain | #Removed | Pruning Ratio | Removed Layer IDs (0-based) |
|--------|----------|---------------|------------------------------|
| 35 | 1 | 2.778% | [32] |
| 34 | 2 | 5.556% | [32, 31] |
| 33 | 3 | 8.333% | [32, 31, 30] |
| 32 | 4 | 11.111% | [32, 31, 30, 2] |
| 31 | 5 | 13.889% | [32, 31, 30, 2, 29] |
| 30 | 6 | 16.667% | [32, 31, 30, 2, 29, 26] |
| 29 | 7 | 19.444% | [32, 31, 30, 2, 29, 26, 1] |
| 28 | 8 | 22.222% | [32, 31, 30, 2, 29, 26, 1, 28] |
| 27 | 9 | 25.000% | [32, 31, 30, 2, 29, 26, 1, 28, 27] |
| 26 | 10 | 27.778% | [32, 31, 30, 2, 29, 26, 1, 28, 27, 25] |
| 25 | 11 | 30.556% | [32, 31, 30, 2, 29, 26, 1, 28, 27, 25, 24] |
| 24 | 12 | 33.333% | [32, 31, 30, 2, 29, 26, 1, 28, 27, 25, 24, 20] |
| 23 | 13 | 36.111% | [32, 31, 30, 2, 29, 26, 1, 28, 27, 25, 24, 20, 19] |
| 22 | 14 | 38.889% | [32, 31, 30, 2, 29, 26, 1, 28, 27, 25, 24, 20, 19, 7] |
| 21 | 15 | 41.667% | [32, 31, 30, 2, 29, 26, 1, 28, 27, 25, 24, 20, 19, 7, 18] |
| 20 | 16 | 44.444% | [32, 31, 30, 2, 29, 26, 1, 28, 27, 25, 24, 20, 19, 7, 18, 8] |
| 19 | 17 | 47.222% | [32, 31, 30, 2, 29, 26, 1, 28, 27, 25, 24, 20, 19, 7, 18, 8, 17] |
| 18 | 18 | 50.000% | [32, 31, 30, 2, 29, 26, 1, 28, 27, 25, 24, 20, 19, 7, 18, 8, 17, 21] |
| 17 | 19 | 52.778% | [32, 31, 30, 2, 29, 26, 1, 28, 27, 25, 24, 20, 19, 7, 18, 8, 17, 21, 22] |
| 16 | 20 | 55.556% | [32, 31, 30, 2, 29, 26, 1, 28, 27, 25, 24, 20, 19, 7, 18, 8, 17, 21, 22, 23] |

*Table 8.* Robustness of the estimated transition on Llama3-8B/ARC-Challenge under evaluation subsets, prompt variants, and option-order perturbations. The transition exists consistently before collapse and disappears consistently after collapse.

| Setting | Pruning | Acc mean/std | T exist rate | T idx mean/std | Final DM mean/std |
|---|---|---|---|---|---|
| Subsets | Dense | 0.83/0.0107 | 1 | 18/0 | 2.9228/0.0993 |
| | 10% | 0.82/0.0084 | 1 | 18/0 | 1.9787/0.0655 |
| | 20% | 0.77/0.0097 | 1 | 18/0 | 1.0112/0.0536 |
| | 30% | 0.80/0.0099 | 1 | 18/0 | 2.0678/0.0927 |
| | 40% | 0.28/0.0054 | 0 | – | -0.7497/0.0123 |
| | 50% | 0.24/0.0109 | 0 | – | -1.0068/0.0387 |
| Prompt variants | Dense | 0.83/0.0059 | 1 | 18/0 | 2.7839/0.1482 |
| | 10% | 0.82/0.0099 | 1 | 18/0 | 1.7659/0.1816 |
| | 20% | 0.74/0.0224 | 1 | 18/0 | 0.8964/0.0695 |
| | 30% | 0.78/0.0238 | 1 | 18/0 | 2.1089/0.1861 |
| | 40% | 0.28/0.0047 | 0 | – | -0.7708/0.0686 |
| | 50% | 0.25/0.0127 | 0 | – | -0.9760/0.0597 |
| Option-order | Dense | 0.838/0.0100 | 1 | 18/0 | 2.9861/0.0035 |
| | 10% | 0.832/0.0020 | 1 | 18/0 | 2.1011/0.0798 |
| | 20% | 0.793/0.0190 | 1 | 18/0 | 1.0551/0.0609 |
| | 30% | 0.816/0.0020 | 1 | 18/0 | 2.1629/0.0981 |
| | 40% | 0.270/0.0100 | 0 | – | -0.7425/0.0113 |
| | 50% | 0.230/0.0100 | 0 | – | -1.0562/0.0052 |

*Table 9.* Transition layers on MMLU math and physics subsets. The transition location is task- and domain-sensitive, rather than a fixed universal layer across all datasets.

| Model | Dataset | Transition layer |
|---|---|---|
| Llama3-8B | H-Math | No transition |
| | C-Math | No transition |
| | H-Phy | 30 |
| | C-Phy | 25 |
| Qwen3-4B | H-Math | 26 |
| | C-Math | 26 |
| | H-Phy | 24 |
| | C-Phy | 24 |

*Table 10.* Entropy-based control for option-frequency bias on Llama3-8B/ARC-Challenge. Entropy is computed from the OF distribution at representative layers. The pattern is stable across subsets, prompts, and option-order shuffling, suggesting that OF dynamics are not mainly driven by superficial option-position bias.

| Setting | Pruning | Mid-Silent (8) | Transition (17) | Late-Decisive (31) |
|---|---|---|---|---|
| Evaluation subsets | Dense | $0.9033 \pm 0.0029$ | $1.3817 \pm 0.0024$ | $1.3821 \pm 0.0022$ |
| | 10% | $0.9033 \pm 0.0029$ | $1.3817 \pm 0.0024$ | $1.3689 \pm 0.0052$ |
| | 50% | $1.1155 \pm 0.0044$ | $1.1831 \pm 0.0021$ | $1.1831 \pm 0.0021$ |
| Prompt variants | Dense | $0.8837 \pm 0.0571$ | $1.3766 \pm 0.0005$ | $1.3804 \pm 0.0008$ |
| | 10% | $0.8837 \pm 0.0571$ | $1.3766 \pm 0.0005$ | $1.3601 \pm 0.0058$ |
| | 50% | $0.8440 \pm 0.0062$ | Removed | $1.1985 \pm 0.0009$ |
| Option-order shuffle | Dense | $0.9038 \pm 0.0007$ | $1.3810 \pm 0.0036$ | $1.3820 \pm 0.0041$ |
| | 10% | $0.9038 \pm 0.0007$ | $1.3810 \pm 0.0036$ | $1.3680 \pm 0.0093$ |
| | 50% | $1.1201 \pm 0.0000$ | Removed | $1.1857 \pm 0.0017$ |

*Table 11.* Lens-dependence control on Llama3-8B/ARC-Challenge. Different decoding lenses change absolute OF statistics, but preserve the transition layer and yield high top-1 agreement.

| Lens | T idx | Acc |
|------|-------|-----|
| Raw | 18 | 0.826 |
| Norm | 18 | 0.828 |
| Affine | 18 | 0.820 |

| Agreement | Rate |
|-----------|------|
| Raw vs. Norm | 0.998 |
| Raw vs. Affine | 0.932 |
| Norm vs. Affine | 0.930 |

*Table 12.* Three-layer sliding-window deletion on Llama3-8B. Windows in the Silent and transition-supporting regions cause much larger accuracy degradation than windows in the Decisive phase, supporting the phase-dependent criticality of the transition scaffold.

| Task | Early Silent (0,1,2) | Mid Silent (8,9,10) | Late Silent (15,16,17) | Transition (16,17,18) | Early Decisive (17,18,19) | Mid Decisive (23,24,25) | Late Decisive (29,30,31) |
|------|-----------|-----------|-----------|-----------|-----------|-----------|-----------|
| ARC-C | 0.232 | 0.510 | 0.346 | 0.364 | 0.836 | 0.830 | 0.840 |
| HellaS | 0.232 | 0.270 | 0.450 | 0.412 | 0.510 | 0.716 | 0.744 |
| ARC-E | 0.236 | 0.632 | 0.624 | 0.472 | 0.786 | 0.878 | 0.898 |

*Table 13.* Comparative zero-shot performance (%) of our proposed Iterative Pruning (IP) against state-of-the-art baselines (ShortGPT, SLEB, and MKA) after Supervised Fine-Tuning (SFT) on Llama3-8B. IP consistently achieves the highest retention of accuracy by adaptively preserving the model's decision scaffolding.

| Model | Method | Ratio | MMLU | BoolQ | Arc-E | Arc-C | PIQA | WingoG | HellaS | Avg |
|-------|--------|-------|------|-------|-------|-------|------|--------|--------|-----|
| | ShortGPT | | 31.19 | 89.85 | 88.09 | 77.05 | 49.89 | 49.57 | 64.88 | 64.36 |
| | SLEB | | 54.69 | 89.60 | 81.23 | 72.35 | 50.49 | 49.57 | 25.25 | 60.46 |
| | MKA | 12.50% | 55.95 | 37.83 | 82.95 | 73.04 | 49.78 | 49.64 | 69.04 | 59.75 |
| | IT-Prun | | 58.84 | 89.64 | 90.48 | 76.96 | 88.90 | 86.50 | 94.97 | **83.76** |
| | ShortGPT | | 46.23 | 90.03 | 87.54 | 77.82 | 51.41 | 49.72 | 24.85 | 61.09 |
| | SLEB | | 38.12 | 62.17 | 72.39 | 56.74 | 78.40 | 50.43 | 25.39 | 54.81 |
| | MKA | 21.90% | 52.46 | 86.15 | 79.12 | 73.12 | 50.71 | 49.57 | 25.58 | 59.53 |
| Llama3-8B | IT-Prun | | 58.44 | 89.18 | 89.85 | 76.02 | 89.23 | 87.21 | 94.48 | **83.49** |
| | ShortGPT | | 59.78 | 89.14 | 86.53 | 75.34 | 50.76 | 49.57 | 29.35 | 62.93 |
| | SLEB | | 40.17 | 86.76 | 46.51 | 36.69 | 64.31 | 51.22 | 25.78 | 50.21 |
| | MKA | 31.20% | 54.72 | 62.17 | 79.34 | 69.71 | 49.51 | 49.72 | 26.89 | 56.01 |
| | IT-Prun | | 61.93 | 89.66 | 89.68 | 79.86 | 88.03 | 88.16 | 94.56 | **84.56** |

*Table 14.* Zero-shot comparison between one-shot ShortGPT-style pruning and IT-Prun. Results are reported without recovery or SFT. The comparison is mixed: ShortGPT is stronger at some low pruning ratios on Llama3-8B, whereas IT-Prun delays collapse or improves retention in other settings.

| Model | Method | Ratio | ARC-C | HellaS | ARC-E | Avg | Removed layers at 30% |
|---|---|---|---|---|---|---|---|
| Llama3-8B | ShortGPT | 10% | 82.25 | 70.74 | 92.38 | 81.79 | – |
| | IT-Prun | 10% | 75.26 | 51.17 | 86.69 | 71.04 | – |
| | ShortGPT | 20% | 82.25 | 68.11 | 92.34 | 80.90 | – |
| | IT-Prun | 20% | 67.15 | 33.12 | 78.82 | 59.70 | – |
| | ShortGPT | 30% | 45.39 | 28.68 | 54.78 | 42.95 | [24,23,25,26,27,28,22,21,29,19] |
| | IT-Prun | 30% | 75.51 | 62.46 | 84.76 | 74.24 | [24,23,25,26,27,28,22,21,19,20] |
| Qwen3-4B | ShortGPT | 10% | 54.31 | 62.88 | 71.87 | 63.02 | – |
| | IT-Prun | 10% | 79.77 | 87.88 | 94.61 | 87.42 | – |
| | ShortGPT | 20% | 27.24 | 23.55 | 27.16 | 25.98 | – |
| | IT-Prun | 20% | 34.51 | 57.59 | 73.81 | 55.30 | – |
| | ShortGPT | 30% | 24.94 | 21.93 | 24.00 | 23.62 | – |
| | IT-Prun | 30% | 25.12 | 22.44 | 23.92 | 23.83 | – |

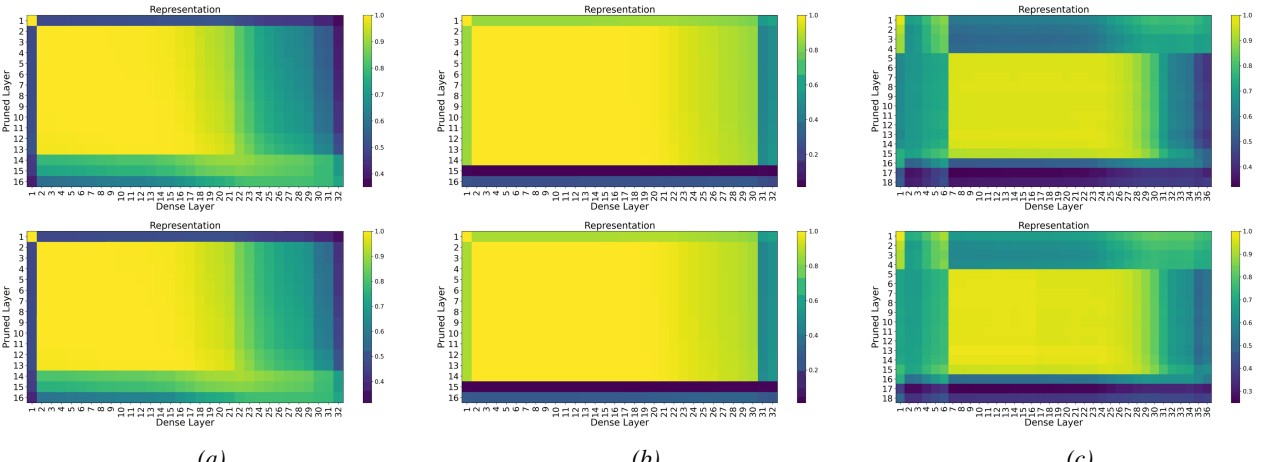

|  (a)  |  (b)  |  (c)  |

*Figure 10.* CKA similarity heatmaps between the hidden representations of dense and 50%-pruned models on ARC-Challenge (Top) and ARC-Easy (Bottom). Lighter regions indicate high representational alignment. The results across (a) Llama3-8B, (b) Llama2-7B, and (c) Qwen3-4B confirm that even collapsed models retain deep-layer semantic features.

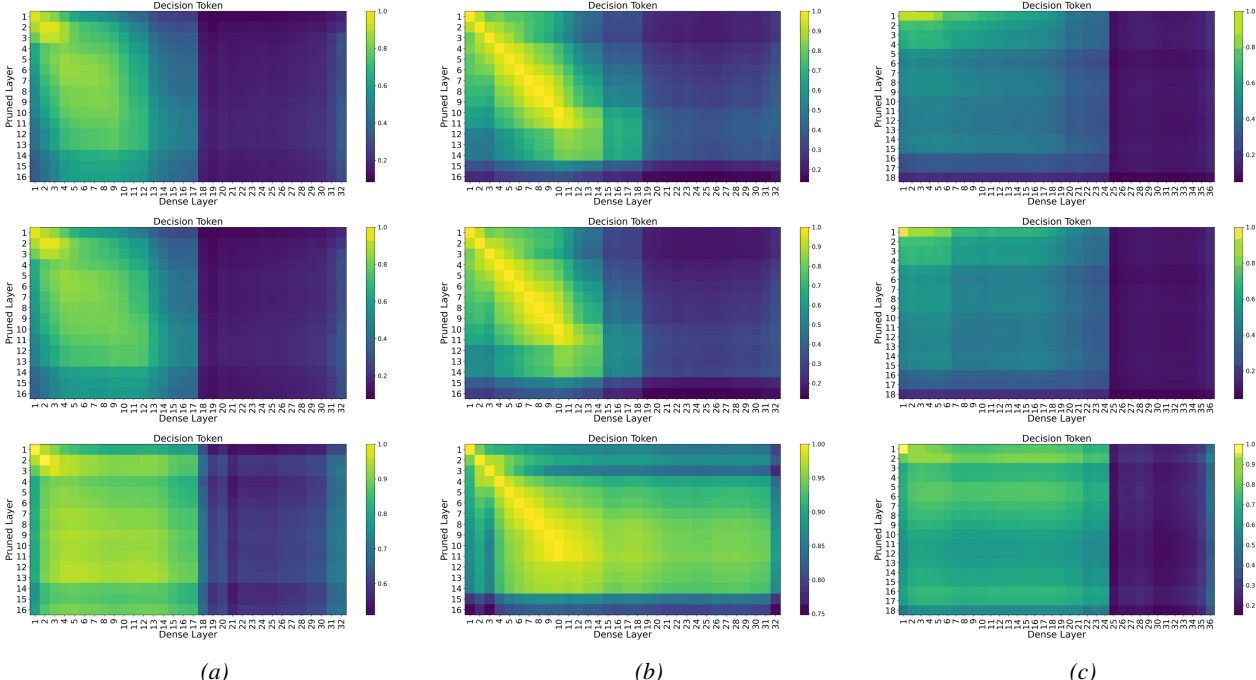

*(a)*                                      *(b)*                                      *(c)*

*Figure 11.* CKA similarity heatmaps focusing on decision-token representations across ARC-Challenge (Top), ARC-Easy (Middle), and Hellaswag (Bottom). The dark regions in the right quadrants highlight the structural truncation of deep decision semantics in heavily pruned (50%) models.

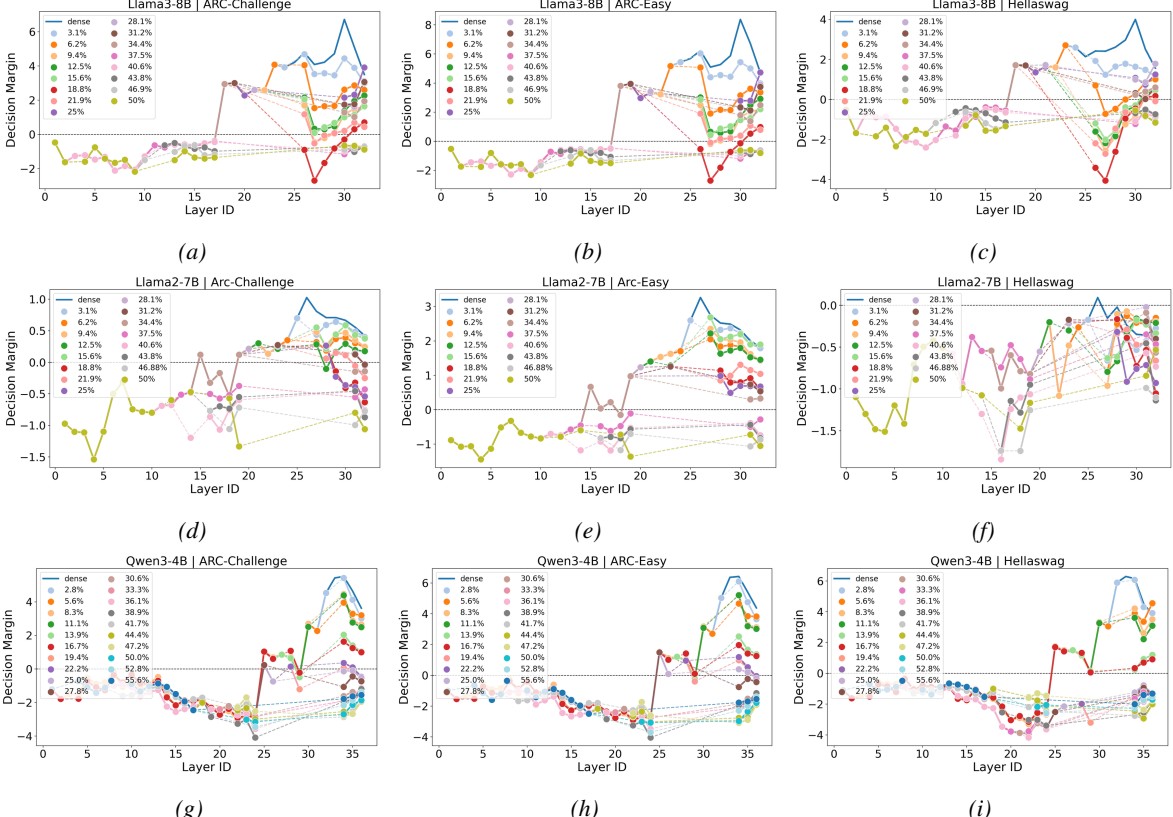

*(a)*                                      *(b)*                                      *(c)*

*(d)*                                      *(e)*                                      *(f)*

*(g)*                                      *(h)*                                      *(i)*

*Figure 12.* Decision Margin (DM) evolution across the complete range of pruning ratios and tasks. Solid lines denote the indices of retained layers mapped back to the original dense model, while dashed lines indicate pruned regions. Performance collapse is consistently triggered when pruning disrupts the Silent-to-Decisive transition boundary.

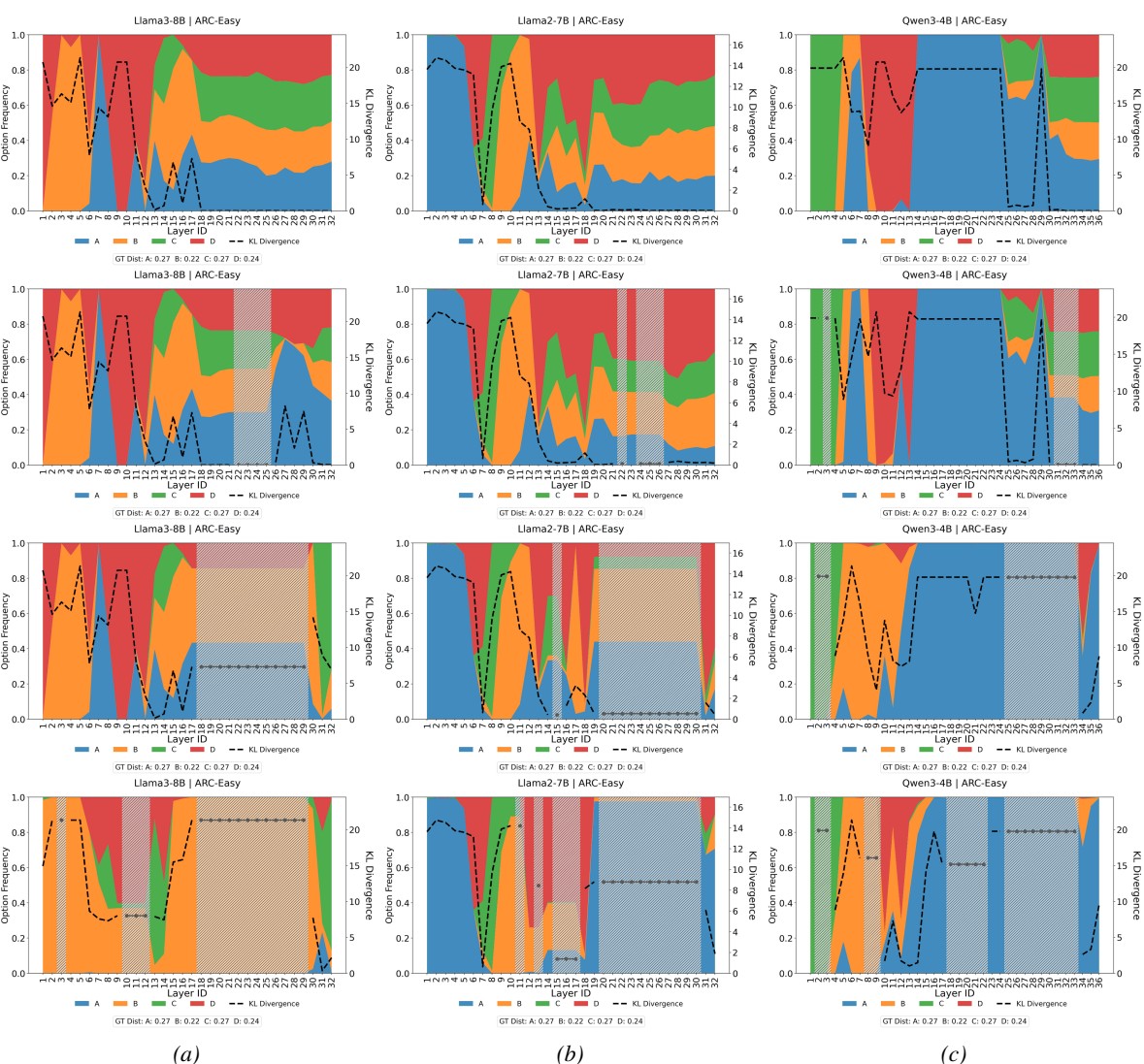

*Figure 13.* Layer-wise KL divergence and option-argmax distributions on the ARC-Easy task under progressive pruning. The sequence from Line 1 (Dense) to Line 4 (50% Pruned) tracks how the model's ability to resolve initial prediction bias is progressively disabled as layers are removed from the Silent Phase. Among them, Line2 is pruned models without performance collapse (Pruning Ratio (PR)=12.5% for Llama3-8B/Llama2-7B, PR=11.1% for Qwen3-4B). Line3 is pruned models with performance collapse (PR=37.5% for Llama3-8B/Llama2-7B, PR=30.56% for Qwen3-4B)

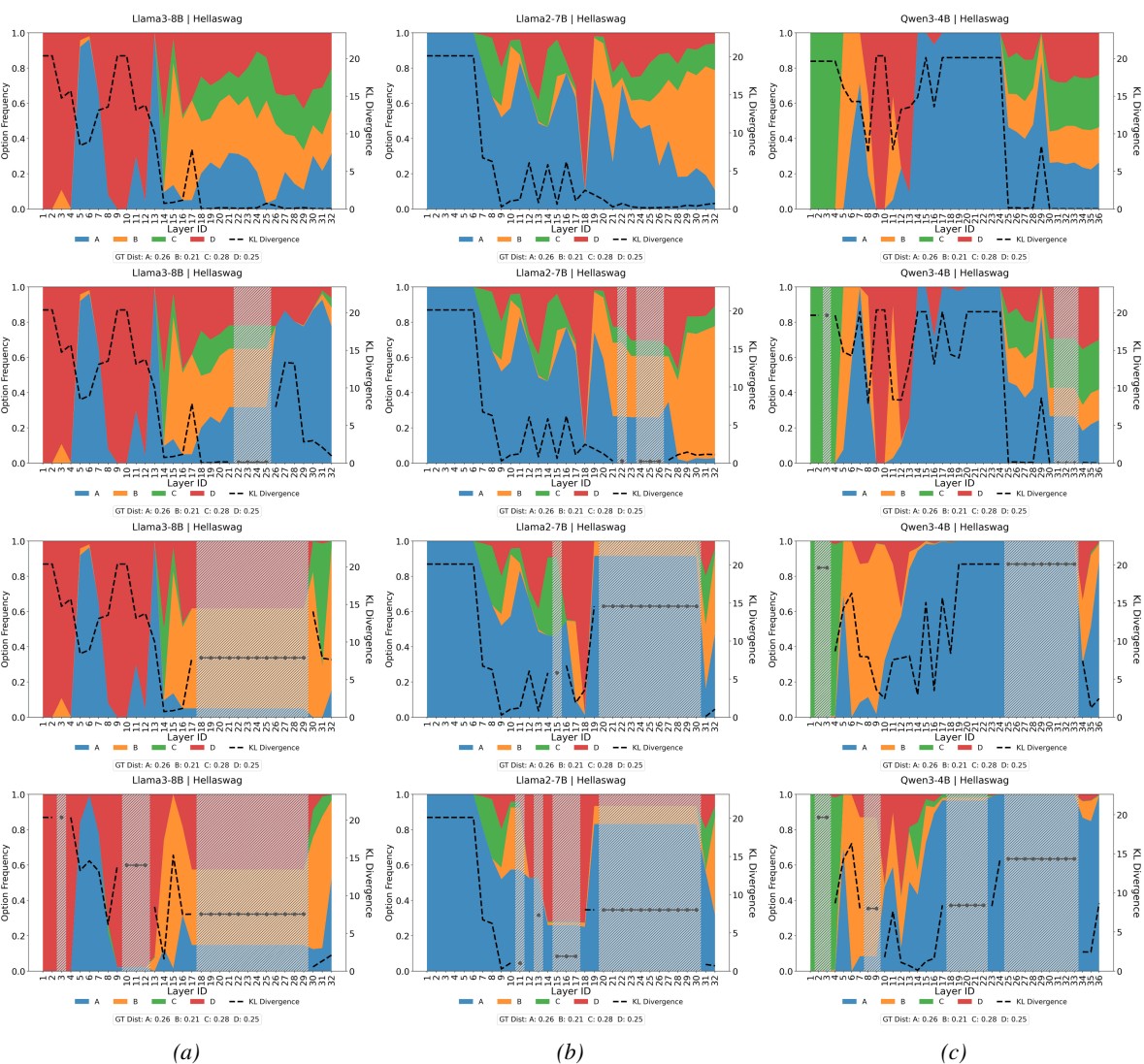

*Figure 14.* Layer-wise KL divergence and option-argmax distributions on the Hellaswag task. Compared to ARC tasks, the distributions here exhibit higher initial instability, yet the fundamental failure of heavily pruned models (Line 4) to recover from early-layer bias remains consistent.

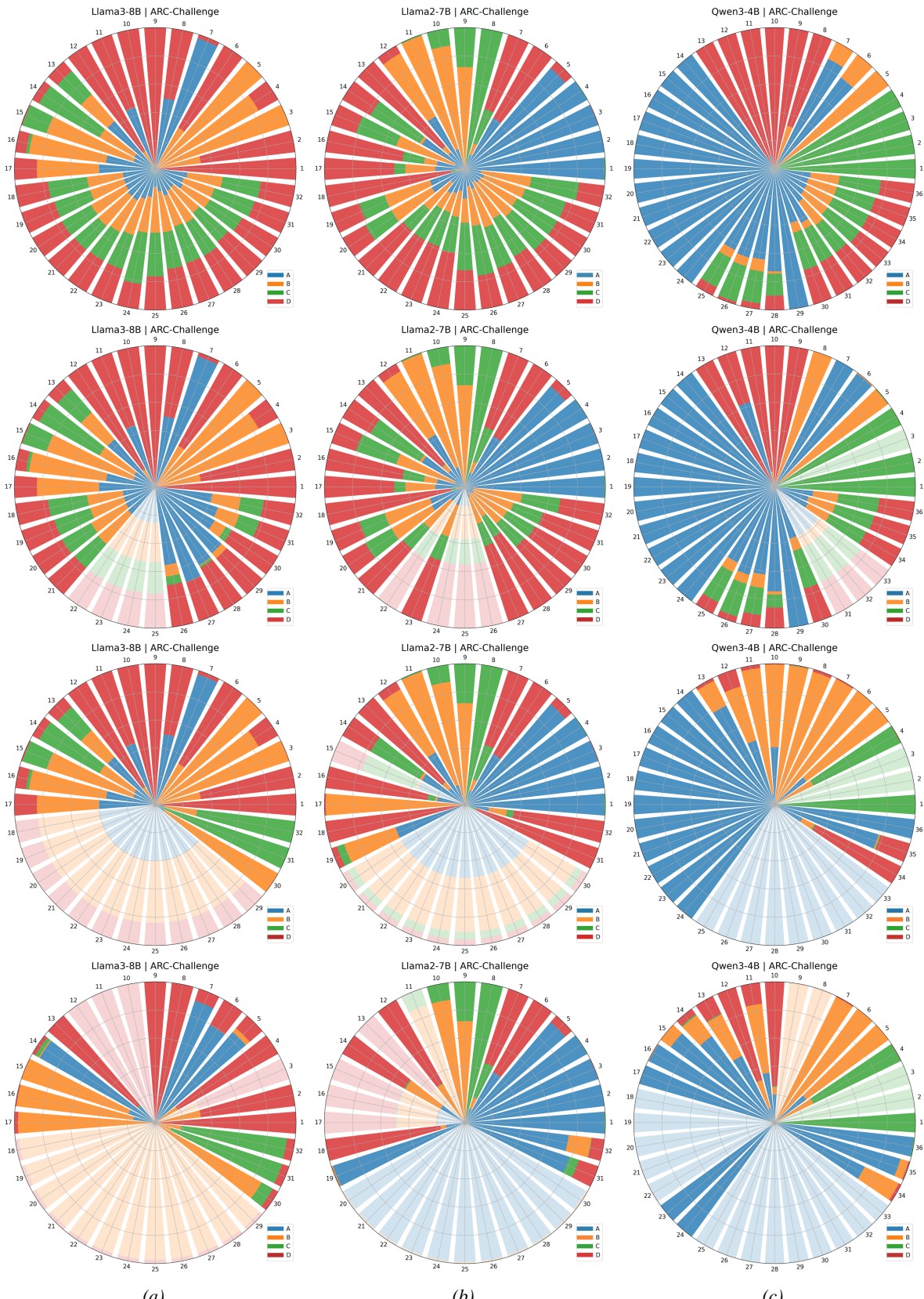

*Figure 15.* Radar visualizations of layer-wise option-argmax distributions on ARC-Challenge. These plots illustrate the expansion of prediction mass: stable models expand from a biased distribution to the correct option, while collapsed models (Line 4) remain trapped in a narrow, biased state. The pruning ratios in each line are the same as in Figure 13.

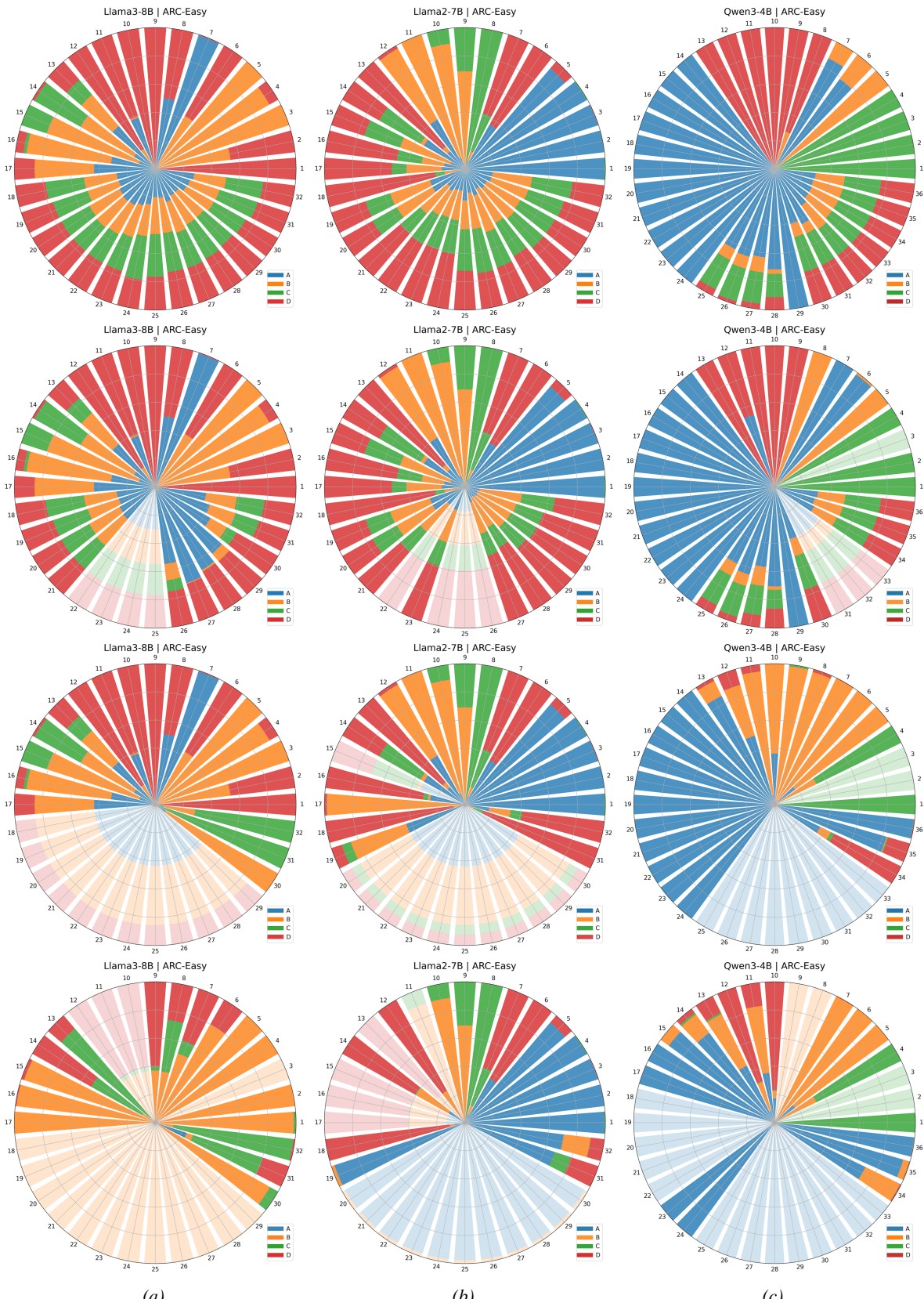

*Figure 16.* Radar visualizations of option-argmax distributions on ARC-Easy. The discrete axes represent candidate options, showing how pruning the Decisive Phase maintains the model's ability to converge on the correct answer, whereas Silent-Phase pruning leads to distribution collapse.

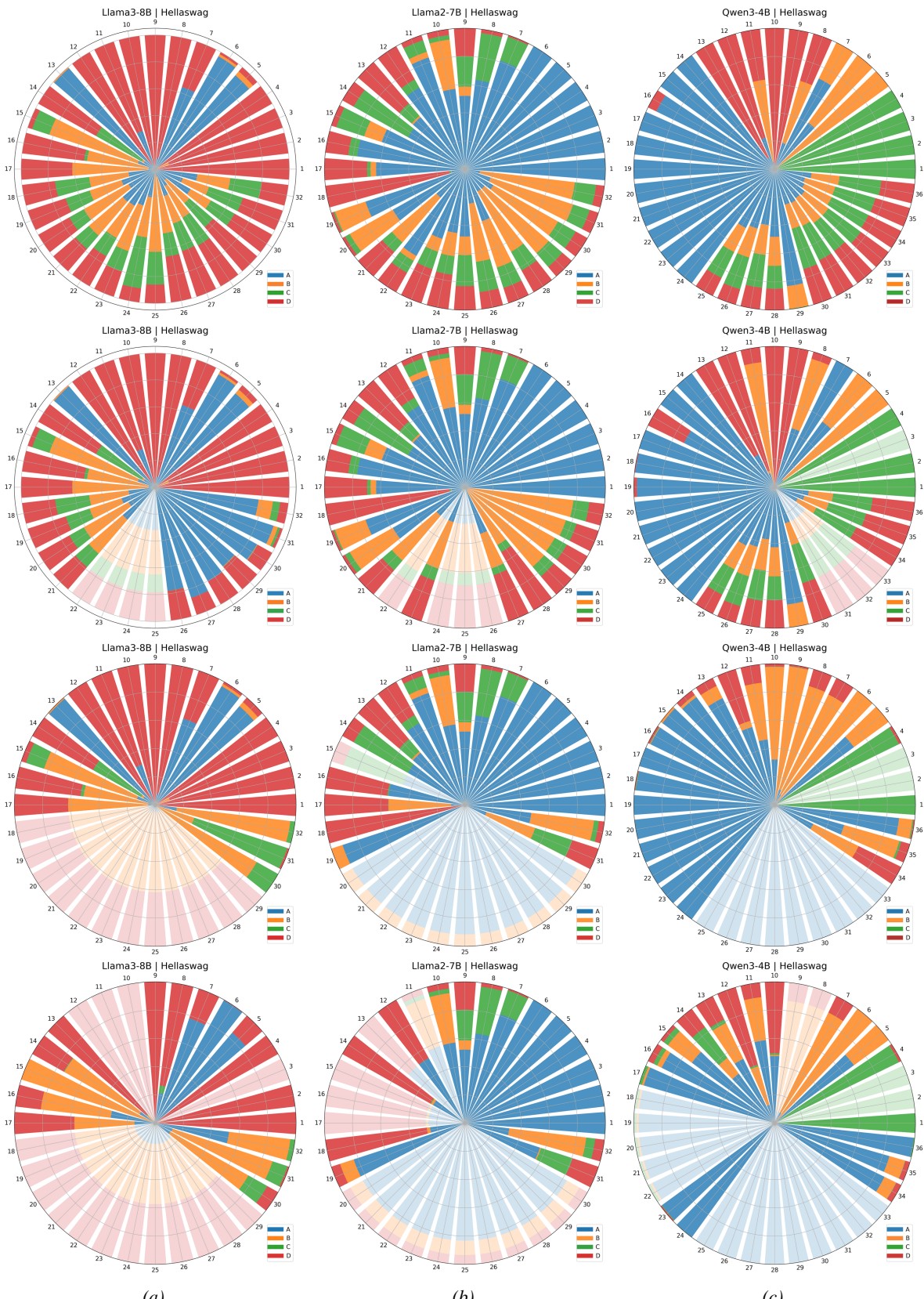

*Figure 17.* Radar visualizations of option-argmax distributions on Hellaswag. The multi-axial plots highlight the persistent prediction bias in Llama2-7B and Llama3-8B when pruning exceeds critical structural thresholds.

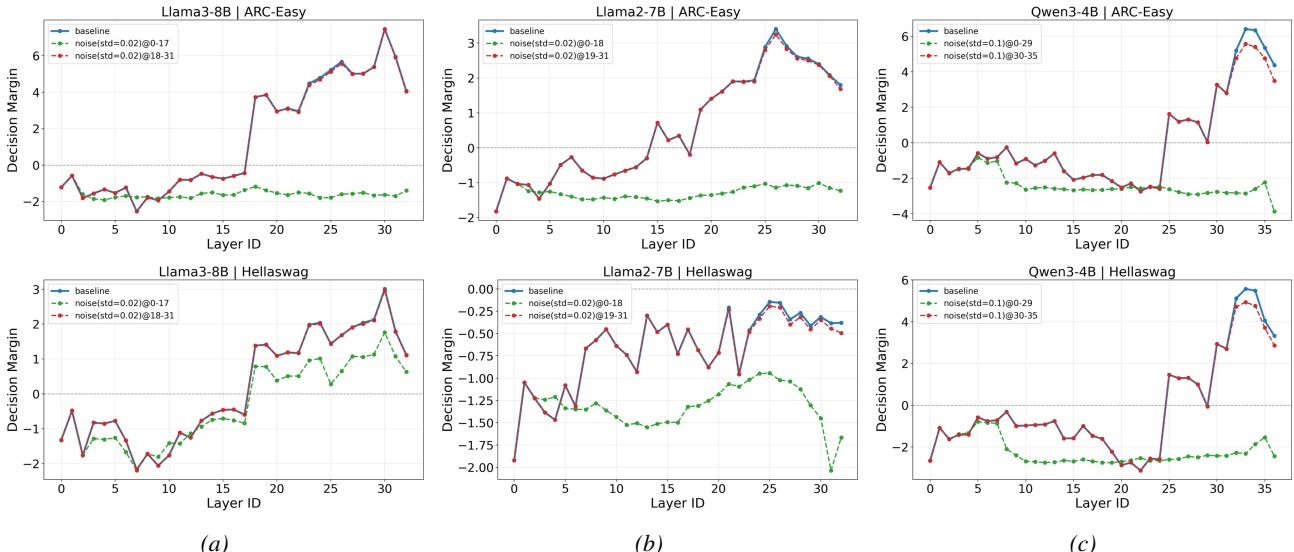

*Figure 18.* Phase-dependent noise sensitivity of the Decision Margin on ARC-Easy (Top) and Hellaswag (Bottom). The significant fluctuations in the Silent Phase across all models reinforce its role as a fragile structural scaffold, in contrast to the high error-tolerance of the Decisive Phase.

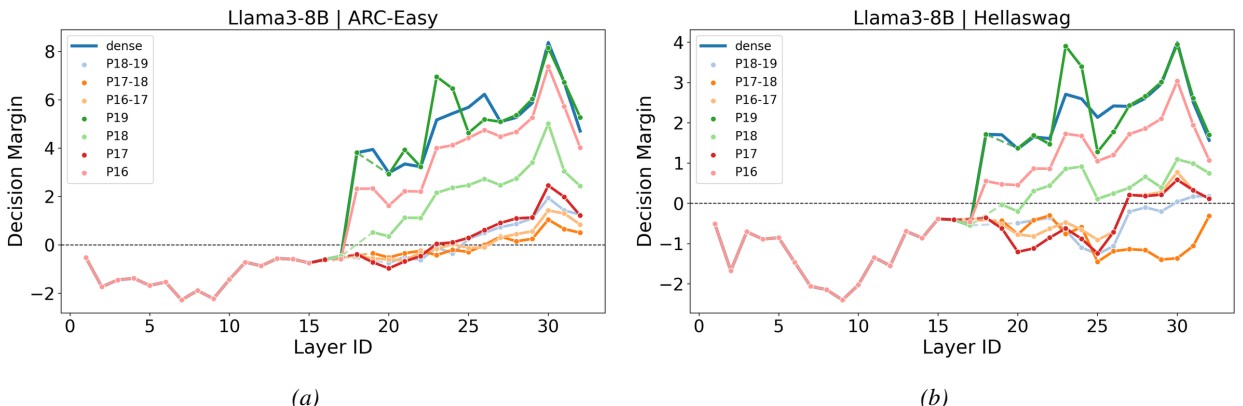

*Figure 19.* Ablation analysis of the decision transition point for Llama3-8B on ARC-Easy and Hellaswag. Targeted removal of individual or paired layers within the transition boundary (Layers 17–18) induces a significantly sharper decline in discriminative capacity than removing deep-layer counterparts.

