# OpenReview forum: "Understanding Performance Collapse in Layer-Pruned Large Language Models via Decision Representation Transitions"
_ICML.cc/2026/Conference — ICML 2026 regular_

### Official Review · Reviewer_RD6A · 2026-02-28

**Soundness:** 2
**Presentation:** 3
**Significance:** 3
**Originality:** 2
**Overall Recommendation:** 4
**Confidence:** 4

**Summary:**

This paper investigates why layer-pruned LLMs exhibit abrupt performance collapse rather than smooth degradation. The authors propose a decision-centered analysis using Decision Margin and Option Frequency to track how answer selection emerges across layers in multiple-choice tasks. They argue that models pass through a Silent Phase followed by a Decisive Phase, and show that iterative layer pruning gradually disrupts this transition.

**Compliance With Llm Reviewing Policy:**

Affirmed.

**Key Questions For Authors:**

**Q1:** The caption of Figure 3 appears inconsistent with the figure content.

**Q2:** In Figure 4, the “DM jump” appears to occur at the same layer for different tasks within the same model. I am curious whether this is a universal phenomenon across tasks. Could the authors provide results on datasets from different domains (e.g., math or physics) to broaden the evidence?

**Q3:** Regarding the OF term, different models appear to show different preferences over selected options. Is this behavior potentially biased by the decoding tool used (e.g., LogitLens), or is there a deeper explanation for this phenomenon?

**Limitations:**

yes

**Strengths And Weaknesses:**

### Strengths

**S1:** The decision-centric framing is interesting and yields a coherent empirical narrative.

**S2:** The contrast between preserved hidden-state similarity and disrupted decision formation provides the central motivation of the paper, making its argument well-established and logically coherent.

**S3:** The paper evaluates multiple model families across several multiple-choice benchmarks. The transition-point analysis, OF/KL dynamics, noise sensitivity, targeted layer ablations, and SFT recovery experiments collectively strengthen the empirical evidence and make the study more convincing.

------

### Weaknesses



**W1:** The practical implications remain somewhat unclear. The paper argues that the discovered phase structure can guide "principled transition-aware pruning" (Line 437), yet it is vague what concrete algorithmic principle this yields beyond the Iterative Pruning analysis itself. In particular, if the recommendation is to preserve the Silent/transition region and prune the Decisive Phase first, the paper should clarify how its results differ from the BI-based Iterative Pruning already used to derive the conclusion. Ideally, it should demonstrate a concrete pruning method built on this principle.

**W2:** The entire framework depends on discrete candidate options and ground-truth labels. While this makes the analysis clean, it also limits the scope of the conclusions. It remains unclear whether the same “Silent Phase / Decisive Phase” phenomenon holds for open-ended generation tasks where there is no fixed option set.

**W3:** The OF metric may be confounded by dataset label statistics, such as option imbalance in the dataset or answer-position bias introduced by prompt templates. A more rigorous analysis should be included to support the claim that OF reflects intrinsic decision dynamics rather than superficial biases.

---

> ### Author Rebuttal · Authors · 2026-03-31
>
> Dear Reviewer RD6A,
>
> Thank you for your constructive comments. We address them below:
>
> > **W1: The pruning implication is still not concrete enough algorithmically.**
>
> We add a simple **Transition-Aware Pruning (TAP)** baseline. TAP is not a replacement for BI, but rather a phase-constrained BI rule: identify the dense model’s transition index $T$, protect a narrow band around it (here, $[T-1, T, T+1]$), and apply the same BI ranking only to the remaining layers. Thus, BI is unconstrained importance-based pruning, whereas TAP applies BI under a transition-aware structural constraint.
>
> This makes the pruning implication algorithmically concrete. TAP matches BI at low pruning ratios, but is more robust at medium/high pruning ratios because it better preserves the transition-supporting region (layers 16–18). Representative results are shown below.
>
> |Llama3-8B|Prune Ratio|BI Acc|TAP Acc|
> |:-:|:-:|:-:|:-:|
> |ARC-Challenge|0|0.81|0.81|
> ||10|0.77|0.77|
> ||20|0.68|**0.76**|
> ||40|0.28|**0.41**|
> |HellaSwag|0|0.76|0.76|
> ||20|0.45|**0.58**|
> ||40|0.29|0.21|
>
> > **W2: The framework’s validity beyond multiple-choice tasks is unclear.**
>
> To test whether the phenomenon is confined to multiple-choice settings, we further evaluate the phenomenon on GSM8K, an open-ended generation benchmark. Detailed results are included in our Response to Reviewer g25d on W1.
>
> > **W3: OF may be influenced by superficial option biases.**
>
> We test this on Llama3-8B/ARC-Challenge under three perturbation settings: different evaluation subsets, prompt templates, and seeds for option-order shuffling. We summarize each layer’s OF distribution by entropy:
> $$
> H_l=-\sum_k \bar p_l(k)\log \bar p_l(k),\quad
> \bar p_l(k)=\frac{1}{N}\sum_i p_{i,l}(k),
> $$
> where $p_{i,l}(k)$ is the probability of option $k$ for sample $i$ at layer $l$. Entropy is compact and **invariant to option renaming and permutation**, so it is suitable for testing positional bias in answer preference. Lower entropy indicates stronger concentration on a few options, while higher entropy indicates a more uniform option distribution.
>
> Across all three perturbations, the entropy pattern remains consistent: the Silent and transition regions stay at low entropy, indicating concentration on a few options. In contrast, the Decisive region has high entropy before collapse (10% pruning ratio) but also falls to low entropy after collapse (50% pruning ratio). This shows that OF is not mainly driven by superficial option-level bias.
>
> |Setting|Prune Ratio|Mid-Silent(8)|Transition(17)|Late-Decisive(31)|
> |:-:|:-:|:-:|:-:|:-:|
> |Evaluation subsets|dense|0.9033±0.0029|1.3817±0.0024|1.3821±0.0022|
> ||10|0.9033±0.0029|1.3817±0.0024|1.3689±0.0052|
> ||50|1.1155±0.0044|1.1831±0.0021|1.1831±0.0021|
> |Prompt variants|dense|0.8837±0.0571|1.3766±0.0005|1.3804±0.0008|
> ||10|0.8837±0.0571|1.3766±0.0005|1.3601±0.0058|
> ||50|0.8440±0.0062|REMOVED|1.1985±0.0009|
> |Option-order shuffle|dense|0.9038±0.0007|1.3810±0.0036|1.3820±0.0041|
> ||10|0.9038±0.0007|1.3810±0.0036|1.3680±0.0093|
> ||50|1.1201±0.0000|REMOVED|1.1857±0.0017|
>
> > **Q1: Figure 3’s caption seems mismatched with the figure.**
>
> We thank the reviewer for pointing this out. We will revise the caption to clarify that, under heavy pruning, upper-layer features may still resemble those of the dense model even though decision behavior remains confined to lower-level representations.
>
> > **Q2: It is unclear whether the DM jump layer generalizes across domains.**
>
> We further test **Llama3-8B** and **Qwen3-4B** on **math** (MMLU college/high-school mathematics: C-Math, H-Math) and **physics** (MMLU college/high-school physics: C-Phy, H-Phy) tasks. The DM transition pattern generalizes beyond commonsense benchmarks, but it is not fixed at exactly the same layer across all tasks. Instead, it is **task-sensitive**: tasks within the same domain tend to show transitions at nearby layers, while different domains can shift the location of the transition.
>
> |Model|Dataset|Transition Layer in Llama3-8B|
> |:-:|:-|:-:|
> |Llama3-8B|H-Math|No Transition|
> ||C-Math|No Transition|
> ||H-Phy|30|
> ||C-Phy|25|
> |Qwen3-4B|H-Math|26|
> ||C-Math|26|
> ||H-Phy|24|
> ||C-Phy|24|
>
> > **Q3: OF differences may depend on the decoding lens.**
>
> We evaluate lens dependence on Llama3-8B/ARC-Challenge using **raw LogitLens**, norm lens, and **affine lens**. Top-1 agreement denotes the fraction of samples for which two lenses produce the same top-1 option at the final layer. Although the decoding lens changes the absolute OF statistics, all three lenses detect the same transition layer, yield very similar final accuracies, and maintain high cross-lens top-1 agreement. Therefore, the observed OF differences are not mainly artifacts of the decoding tool: the lens affects the strength of OF preference, but not the phase behavior or transition location.
>
> |Lens|T_idx|Acc|
> |:-:|:-:|:-:|
> |Raw|18|0.826|
> |Norm|18|0.828|
> |Affine|18|0.820|
>
> |Agreement|Rate|
> |:-:|:-:|
> |Raw vs. Norm|0.998|
> |Raw vs. Affine|0.932|
> |Norm vs. Affine|0.930|

---

> > ### Author Rebuttal · Reviewer_RD6A · 2026-04-01
> >
> > Thanks to the authors’ additional experiments. My concerns have been adequately addressed.

---

> > > ### Author Response · Authors · 2026-04-03
> > >
> > > Thank you for the reviewer's kind Acknowledgement. We are pleased to hear that our previous explanation has answered your questions.

---

### Official Review · Reviewer_jx82 · 2026-03-04

**Soundness:** 3
**Presentation:** 3
**Significance:** 3
**Originality:** 3
**Overall Recommendation:** 4
**Confidence:** 3

**Summary:**

This paper investigates the sudden performance collapse observed in LLMs during layer pruning. Overall, the authors focus on the aspect of decision representations rather than traditional hidden state semantic alignment. The authors introduce two key metrics: Decision Margin (DM) and Option Frequency (OF), along with an Iterative Pruning (IP) method serving as an analytical probe.

The study reveals that LLM inference consists of two distinct stages: a "Silent Phase," where the model has not yet identified the correct answer, and a "Decisive Phase," where reliable predictions emerge. Experiments demonstrate that pruning layers within the Decisive Phase has minimal impact, whereas encroaching on the Silent Phase, particularly disrupting the Transition Point, triggers immediate performance collapse. Through Centered Kernel Alignment (CKA) analysis, the authors show that deep semantic representations remain intact even when performance collapses completely. This challenges the conventional view that collapse is due to the loss of semantic features, arguing instead that it is a structural failure to facilitate the decision transition.

**Compliance With Llm Reviewing Policy:**

Affirmed.

**Final Justification:**

ok

**Key Questions For Authors:**

Does your analysis extend to width pruning as well? In both depth and width pruning, a higher pruning ratio eventually triggers a sharp performance degradation at a specific threshold, although this phenomenon is more evident in depth pruning.

**Limitations:**

yes

**Strengths And Weaknesses:**

## Strength
-  The shift from analyzing "hidden representation similarity" to "decision formation dynamics" is highly compelling. The finding that high CKA similarity can coexist with total functional failure provides a critical correction to existing representation-centric analyses.

- The experiments are comprehensive.

## Weakness
- The current study is primarily confined to multiple-choice tasks, which allow for clear definition of ground truth and DM calculation. It remains unclear how these concepts translate to open-ended generation tasks, limiting the generalizability of the conclusions.

- The paper lacks a clear methodological flowchart to illustrate their approach.

- Many figures are difficult to interpret because their captions fail to clearly explain what the content represents, forcing the reader to guess their meaning. For instance, in Figure 5, it is not specified what labels A, B, C, and D denote. Furthermore, Figure 1 does not explain what the white squares represent, and the significance of the bottom section is unclear.

---

> ### Author Rebuttal · Authors · 2026-03-31
>
> Dear Reviewer jx82,
>
> Thank you for your constructive comments. We address them below:
>
> > **W1: Generalizability beyond multiple-choice tasks.**
>
> To examine whether the transition phenomenon is specific to multiple-choice settings, we further extend our analysis to open generation on GSM8K. The detailed results and discussion are provided in our Response to Reviewer g25d on W1.
>
> > **W2: Missing workflow overview.**
>
> In the revised paper, we will add a workflow figure to summarize the full pipeline, including **dense-model analysis, layer-wise metric computation, transition-layer identification, pruning-path construction, and post-pruning evaluation**. This will make the methodology easier to follow.
>
> > **W3: Figure interpretation needs clarification.**
>
> We will revise the figures and captions to make them self-contained. Specifically, in Fig. 5, we will explicitly state that A/B/C/D denote the four options. In Fig. 1, we will clarify that the white squares indicate pruned layers, and that the bottom panel compares one-shot pruning with iterative pruning. One-shot pruning directly prunes the dense model to the target size, while iterative pruning proceeds from an already pruned model. As iterative pruning may get trapped in local optima, we additionally introduce a correction step that switches to one-shot pruning when such local optima are encountered.
>
> > **Q1: Extension to width pruning.**
>
> Following the reviewer’s suggestion, we also study width pruning using **SliceGPT on Llama3-8B**. The results show that our main observation partially extends to width pruning: **as the pruning ratio increases, both task performance and final DM consistently deteriorate, indicating that width pruning also exhibits a clear threshold-like collapse**. This trend is consistently observed across all evaluated tasks, suggesting that the collapse phenomenon is not unique to depth pruning.
>
> However, width pruning differs from depth pruning in how collapse emerges. In depth pruning, collapse is often associated with a relatively clear transition layer. In width pruning, all layers are retained and only their hidden dimensions are reduced, so the degradation is more distributed across layers. Accordingly, our layer-wise DM curves do not suggest a single layer-localized transition, but rather a more globally distributed loss of confidence.
>
> Therefore, while both pruning types exhibit sharp degradation beyond a threshold, depth pruning reveals a more clearly layer-localized transition structure. Understanding the collapse mechanism of width pruning will be an important direction for future work, and we will investigate this further based on our findings in layer pruning.
>
> | Pruning Ratio | ARC-C Acc | ARC-E Acc | HellaSwag Acc | Mean Acc |
> | :-------------: | :--------: | :--------: |:------------: | :-------: |
> | 10%           |      0.51 |      0.79 |          0.59 |     0.63 |
> | 20%           |      0.48 |      0.73 |          0.57 |     0.59 |
> | 30%           |      0.39 |      0.68 |          0.55 |     0.54 |
> | 40%           |      0.34 |      0.62 |          0.52 |     0.50 |
> | 50%           |      0.32 |      0.58 |          0.48 |     0.46 |
>
> | Pruning Ratio | ARC-C Final DM | ARC-E Final DM | HellaSwag Final DM |
> |:-------------: | :-------------: | :-------------: | :-----------------: |
> | 10%           |         -0.080 |          1.160 |              0.184 |
> | 20%           |         -0.207 |          0.864 |              0.133 |
> | 30%           |         -0.367 |          0.536 |              0.090 |
> | 40%           |         -0.492 |          0.259 |              0.051 |
> | 50%           |         -0.534 |          0.158 |             -0.009 |

---

> > ### Author Rebuttal · Reviewer_jx82 · 2026-04-02
> >
> > ok

---

> > > ### Author Response · Authors · 2026-04-03
> > >
> > > Thank you for the reviewer's kind Acknowledgement. We are pleased to hear that our previous explanation has answered your questions.

---

### Official Review · Reviewer_g25d · 2026-03-13

**Soundness:** 3
**Presentation:** 3
**Significance:** 2
**Originality:** 2
**Overall Recommendation:** 4
**Confidence:** 2

**Summary:**

This paper argues that the sudden collapse observed in LLM layer pruning is not mainly caused by the direct loss of deep semantic representations, but rather by the model’s failure to complete the critical decision transition from the Silent Phase to the Decisive Phase. To characterize this process, the authors introduce DM and OF, and show that the truly fragile part is the early structure that supports decision formation. Once pruning starts to invade this part, the model can no longer reach the transition point, leading to a sharp drop in performance.

**Compliance With Llm Reviewing Policy:**

Affirmed.

**Final Justification:**

### 1. Reproduction of W3 Results

Using the exact block indices and model provided by the authors, I can reproduce the IT-Prun 30% results under their evaluation setting.

But under `lm-evaluation-harness` (v0.4.11, 0-shot), where each option's log-likelihood is computed independently without seeing other options, the degradation is much more severe:

| Task | Dense | IT-Prun 30% | ShortGPT 30% |
|------|-------|-------------|--------------|
| ARC-C | 53.41 | 32.68 | 30.03 |
| ARC-E | 82.03 | 47.43 | 46.63 |
| HellaSwag | 59.81 | 39.22 | 33.04 |
| BoolQ | 85.41 | 62.78 | 43.55 |
| PIQA | 80.20 | 66.70 | 64.09 |
| WinoGrande | 73.56 | 65.51 | 61.17 |
| MMLU | 68.32 | 65.43 | 31.78 |
| **Avg** | **71.82** | **54.25** | **44.33** |

### 2. MC-Only Concern Remains Unresolved

My original concern was that the method is designed specifically for multiple-choice tasks. The rebuttal reinforced rather than resolved this. The evaluation treats the LLM as an ABCD selector with all options visible in the prompt. The GSM8K extension still relies on constructing an answer pool and applying teacher forcing, rather than evaluating actual open-ended generation. The Table 5 fine-tuning setup uses train splits of MC benchmarks (BoolQ, ARC, PIQA, HellaSwag, WinoGrande) as SFT data, further narrowing the model toward MCQ answering. A pruned model that retains the ability to pick the correct letter from visible options but generates incoherent text (e.g., *"The capital of France is a Wessonion..."*) has not meaningfully preserved its capabilities for general use.

### 3. Selective Reporting and Anomalous Results

Looking at the authors' responses across both rounds of W2 and W3, I observe a pattern of selective reporting. In W2, only ARC-C and HellaSwag were reported; in W3, only ARC-C, ARC-E, and HellaSwag — consistently omitting the remaining benchmarks from Table 5 without explanation. These happen to be the tasks where IT-Prun retains the most performance relative to the dense model.

Additionally, some results are anomalous: IT-Prun at 30% pruning (avg=74.24) substantially outperforms IT-Prun at 20% (avg=59.70). A more aggressively pruned model performing much better than a less pruned version is highly unusual. This may reflect artifacts of the evaluation methodology rather than genuine findings, and further weakens my confidence in the experimental claims.

---

**Summary**: I raise my score to **4** as the results are reproducible under the authors' specific evaluation setting, but I lower my confidence to **2** as I remain unconvinced by the overall evaluation methodology and the selective reporting pattern. I respectfully ask the AC to examine the full rebuttal exchange.

**Key Questions For Authors:**

please see weaknesses

**Limitations:**

The biggest limitation is that the entire method is designed specifically for multiple-choice tasks.

**Strengths And Weaknesses:**

Strengths:

- The paper is well written and reads very smoothly.

- The paper identifies an important problem and goes beyond merely describing an empirical phenomenon. Instead, the authors analyze the issue from the perspective of decision boundaries / decision formation.

Weaknesses and Questions:

- The biggest limitation is that the entire method is designed specifically for multiple-choice tasks. In practice, however, people do not use LLMs only for multiple-choice questions. This seriously limits the generalizability of the method and weakens the overall significance of the paper.

- Iterative Pruning has a serious issue: once the k-th block is pruned in the first step, all profiling results for layers after layer k will be affected by the removal of that block. Because of this dependency, a one-shot method may actually be a better choice.

- The paper lacks sufficient comparison with baselines. Analyzing only DM is not enough to demonstrate the superiority of the proposed method. Although Table 5 reports some comparisons, those results are based on fine-tuned models, so I do not think this is a fair comparison.

---

> ### Author Rebuttal · Authors · 2026-03-31
>
> Dear Reviewer g25d,
>
> Thank you for your constructive comments. We address your concerns below:
>
> > **W1 & Limitations: Lacking open-generation tasks.**
>
> Following your suggestion, we extend the analysis to open generation on **GSM8K**. For each sample, we build an answer pool with the gold answer and negatives, apply teacher forcing to score each candidate at each layer, anddefine the layer-wise margin (DM) as the score gap between the gold answer and the strongest negative. This lets us track the same transition statistics as in the MC setting, such as the DM of the final layer (Final DM), the transition-layer index (T Idx), together with Exact Match (EM).
>
> The same trend holds for both Llama3-8B and Qwen3-4B. As pruning grows, EM drops sharply and Final DM degrades, consistent with collapse. This shows that the transition phenomenon is not MC-specific and also appears in open generation.We will revise the paper to clarify that MC is used as a controlled setting for mechanistic analysis, while the transition dynamics generalize beyond MC.
>
> |Ratio%|EM|Final DM|T Idx|
> |:-:|:-:|:-:|:-:|
> |Dense|0.66|0.2365|23|
> |10|0.37|0.0086|23|
> |20|0.00|-0.1674|—|
> |30|0.00|-0.2166|—|
> |40|0.00|-0.3064|—|
> |50|0.00|-0.4510|—|
>
> |Ratio%|EM|Final DM|T Idx|
> |:-:|:-:|:-:|:-:|
> |Dense|0.74|0.7486|29|
> |10|0.68|0.4645|29|
> |20|0.01|-0.0862|—|
> |30|0.00|-0.4285|—|
> |40|0.00|-0.2366|—|
> |50|0.00|-0.3660|—|
>
> > **W2: The one-shot method may actually be a better choice.**
>
> Iterative pruning introduce inter-step dependencies,since later blocks are evaluated after earlier removals. However, this does not undermine our conclusion. We do not claim that IT-Prun recovers the dense model’s saliency pattern. Rather, we show that transition disappearance aligns with collapse under different pruning protocols.
>
> To verify this, we compare one-shot pruning (ShortGPT-style BI) and iterative pruning (IT-Prun) on Llama3-8B and Qwen3-4B over ARC-Challenge and HellaSwag. Across all settings, collapse occurs when the transition disappears (i.e., when T_exist = 0) and Final DM becomes negative, under both one-shot pruning and IT-Prun. Moreover, IT-Prun delays the onset of collapse in some cases, including Llama3-8B on HellaSwag and Qwen3-4B on both tasks.
>
> One-shot pruning is less flexible, because all pruning decisions are made once from the dense model and cannot be adjusted after earlier removals. In contrast, iterative pruning updates its decisions step by step based on the current pruned model. This makes it better suited for analyzing how decision dynamics evolve along the actual pruning path. Importantly, our conclusion does not rely on IT-Prun alone: the same transition-collapse coupling is observed under both methods.
>
> |Model|Task|Method|Ratio where Trasit disapper|Acc at that ratio|Final DM at that ratio|
> |:-:|:-:|:-:|:-:|:-:|:-:|
> |Llama3-8B|ARC-Challenge|One-shot|40%|0.23|-1.5264|
> |Llama3-8B|ARC-Challenge|IT-Prun|40%|0.28|-0.7250|
> |Llama3-8B|HellaSwag|One-shot|30%|0.27|-1.3205|
> |Llama3-8B|HellaSwag|IT-Prun|40%|0.28|-0.8893|
> |Qwen3-4B|ARC-Challenge|One-shot|10%|0.41|-1.1548|
> |Qwen3-4B|ARC-Challenge|IT-Prun|20%|0.33|-0.5962|
> |Qwen3-4B|HellaSwag|One-shot|10%|0.41|-1.3959|
> |Qwen3-4B|HellaSwag|IT-Prun|20%|0.24|-1.4567|
>
> > **W3: Insufficient baseline.**
>
> To make the comparison more balanced, add a zero-shot comparison between one-shot pruning (ShortGPT) and iterative pruning (IT-Prun). The results are mixed rather than one-sided. On Llama3-8B, ShortGPT is better at the 10% and 20% pruning ratios (81.79/80.90 vs. 71.04/59.70), while IT-Prun performs better at 30% (74.24 vs. 42.95). On Qwen3-4B, IT-Prun is better at 10%/20% (87.42 vs. 63.02, 55.30 vs. 25.98), and their performance is nearly identical at 30% (23.83 vs. 23.62). Therefore, we do **not** claim that IT-Prun is universally superior to ShortGPT. We will revise the paper to make this limitation explicit and avoid drawing overly strong conclusions from the Table 5. The new **zero-shot** comparison provides a fairer and more direct baseline comparison
>
> Overall, our goal is not to show that one pruning method always outperforms another. Rather, the zero-shot results show that simply switching pruning methods yields mixed results, which motivates our main contribution: **identifying the transition-sensitive region around the transition layer**. We further add a simple **Transition-Aware Pruning (TAP)** baseline, which retains the same BI ranking while protecting a narrow band around the dense model’s transition index T, here using [T−1, T, T+1]. In this sense, TAP can be viewed as BI under a transition-aware structural constraint, rather than a new pruning score. We report its better results separately in Response to Reviewer RD6A on W1.
>
>
> |Model|Ratio|ShortGPT Avg|IT-Prun Avg|Δ(IT-Prun-ShortGPT)|
> |:-:|:-:|:-:|:-:|:-:|
> |Llama3-8B|10|81.79|71.04|-10.75|
> ||20|80.90|59.70|-21.20|
> ||30|42.95|74.24|+31.29|
> |Qwen3-4B|10|63.02|87.42|+24.40|
> ||20|25.98|55.30|+29.32|
> ||30|23.62|23.83|+0.21|

---

> > ### Author Rebuttal · Reviewer_g25d · 2026-04-01
> >
> > > W1
> >
> > I appreciate that the authors built an answer pool with the gold answer and negative answers to show that this phenomenon also exists on GSM8K. However, the question is: **when we prune a more general-domain LLM (instead of a model that only answers ABCD choices), do we also need to build such an answer pool? This is clearly unreasonable.** On the other hand, I do not think results on multiple-choice tasks can be directly transferred to general LLMs.
> >
> > > W2
> >
> > I appreciate that the authors ran additional experiments, but I find the results very strange. Why is there no fair and comparable setting?
> >
> > - For Llama3-8B + ARC-C, when the ratio is 40%, the **accuracy is already close to random. Such a comparison is not meaningful.**
> >
> > - For Llama3-8B + HellaSwag, **random-level results** are also meaningless, and the **ratios for One-shot and IT-Prun do not match (30% vs. 40%).** I understand that IT-Prun prunes more aggressively, but **what I want is a fair comparison, not a case where one random result is slightly better than another.**
> >
> > - For Qwen3-4B + ARC-C / HellaSwag, the same issue( 10% vs. 20%).
> >
> > > W3
> >
> > The experimental results are very unusual. For ShortGPT and the method in this paper, both of which prune at the block level given layer indices, **the gap should not be this large unless the comparison is unfair.** At ratio = 30%, under a zero-shot comparison (with no recovery at all), the reported average performance of Llama3-8B is 74.24. I strongly doubt this result.
> >
> > ---
> > # If my understanding is incorrect, please correct me.
> > ---
> > # But if this is indeed the setting I understand: Llama3-8B, pruning ratio = 30%, no recovery, average accuracy = 74.24, then **please provide the block indices. I will test it myself.** If the result is comparable to what you reported, **I will raise my score to 4.**
> >
> > ---
> >
> > In addition, regarding Table 5, it seems the paper does not mention which dataset was used for fine-tuning. This is very tricky. **In the most extreme case, if the test set is used for fine-tuning, the accuracy could even become higher than before.** (I’m not saying or implying anything; I’m just using an extreme example to illustrate the importance of datasets in this kind of scenario, especially for datasets that are not reported.) The quality of the dataset matters a lot. Therefore, the results in Table 5 may not rigorous enough.
> >
> > ---
> >
> > I really appreciate the experimental results provided by the authors, but I find them inconsistent with my basic understanding, especially given that I have conducted some similar experiments myself. I hope to see clarification from the authors.

---

> > > ### Author Response · Authors · 2026-04-02
> > >
> > > Thank you for questions. Space limits force us to compress the tables/explanations, which may have caused misunderstanding. Here, we give detailed analysis.
> > >
> > > > **W1: Whether answer-pool analysis is reasonable beyond MC and transferable to general-domain LLMs**
> > >
> > > Pruning a general-domain LLM with the answer pool is not what we assume. The answer pool in GSM8K is not part of the pruning method. It is introduced only as a post-hoc analysis tool, i.e., a proxy for judging whether the correct generation has become layer-wise dominant, so that decision dynamics remain observable in a generation task.
> > >
> > > Thus, our claim is not that results on MC can be directly transferred to arbitrary free-form generation, nor that general-domain pruning needs such a pool in practice. Our point is that when a reasonable surrogate judgment is available, the same transition/collapse pattern can also be observed beyond standard MC tasks. Extending this analysis to fully open-ended generation without any pool would require a new pool-free decision way, which is beyond this work and an important furture work as noted in our limitations.
> > >
> > > > **W2: Fairness of comparison**
> > >
> > > **The reviewer's understanding of the table in the response to W2 is incorrect**.The table here is not intended to compare the two methods at the same pruning ratio. It aims to compare the pruning ratio at which collapse first occurs. Specifically, “Ratio where Transition disappears” denotes the first pruning ratio where the model loses the transition, and “Acc/Final DM at that ratio” reports the corresponding accuracy and DM at collapse. Thus, the reported ratios for One-shot Prun and IT-Prun may differ, since the question is precisely which method collapses later.
> > >
> > > Moreover, this comparison is fair: all results are zero-shot with no recovery. The results below show the same trend. On Llama3-8B/HellaS, One-shot collapses at 30%, while IT-Prun collapses only at 40%. On Qwen3-4B (ARC-C and HellaS), One-shot collapses at 10%, while IT-Prun collapses at 20%. So our point is not matched-ratio comparison, but that IT-Prun delays collapse.
> > >
> > > |Model|Ratio|ShortGPT|IT-Prun|
> > > |:-:|:-:|:-:|:-:|
> > > |Llama3-8B/ARC-C|0|0.82|0.82|
> > > ||10|0.80|0.77|
> > > ||20|0.81|0.68|
> > > ||30|0.47|0.80|
> > > ||40|0.23|0.28|
> > > ||50|0.21|0.25|
> > > |Llama3-8B/HellaS|0|0.76|0.76|
> > > ||10|0.75|0.56|
> > > ||20|0.70|0.46|
> > > ||30|0.27|0.64|
> > > ||40|0.27|0.28|
> > > ||50|0.20|0.26|
> > > |Qwen3-4B/ARC-C|0|0.88|0.88|
> > > ||10|0.41|0.85|
> > > ||20|0.21|0.33|
> > > ||30|0.21|0.21|
> > > ||40|0.20|0.20|
> > > ||50|0.21|0.21|
> > > |Qwen3-4B/HellaS|0|0.84|0.84|
> > > ||10|0.41|0.80|
> > > ||20|0.20|0.24|
> > > ||30|0.20|0.21|
> > > ||40|0.20|0.20|
> > > ||50|0.20|0.20|
> > >
> > > > **W3:  Unusual experimental results**
> > >
> > > We clarify that the 74.24 reported in Response to W3 is the average over ARC-C, HellaS, and ARC-E. The detailed per-task results are provided in the table below.
> > >
> > > The code is provided in the **Supplementary Material**. To reproduce the results, please download the code and modify `./scripts/llama3-8b-instruct/shortgpt_evaluate_zeroshot.sh` and `itera_shortgpt_evaluate_zeroshot.sh` for shortgpt and IT-Prun as follows:
> > >
> > > * `FINETUNED_MODEL_BASE_PATH`: the root path of the pruned models
> > > * `FINETUNED_MODEL_FILE_LIST`: the parent directory of the pruned models
> > > * `OUTPUT_DIR`: the directory for saving evaluation results
> > >
> > > Please also replace `/TO/MY/PATH/code/` in the `torchrun` command with your own project directory. In addition, you need to update the dataset paths in `./TALE/data_utils.py` (Lines 995–1113) to your local paths.
> > >
> > > For reproduction, we provide the exact removed block indices for ShortGPT and IT-Prun,  plus the evaluation splits in the next response. The models are Meta-Llama-3.1-8B-Instruct and Qwen3-4B-Instruct-2507-modelscope. Models and datasets are from HuggingFace/ModelScope.
> > >
> > > **[Llama3-8B]**
> > >
> > > |Method|Ratio|Arc-C|HellaS|Arc-E|Avg|
> > > |:-:|:-:|:-:|:-:|:-:|:-:|
> > > |ShortGPT|10|82.25|70.74|92.38|**81.79**|
> > > |IT-Prun||75.26|51.17|86.69|71.04|
> > > |ShortGPT|20|82.25|68.11|92.34|**80.90**|
> > > |IT-Prun||67.15|33.12|78.82|59.70|
> > > |ShortGPT|30|45.39|28.68|54.78|42.95|
> > > |IT-Prun||75.51|62.46|84.76|**74.24**|
> > >
> > > **[Qwen3-4B]**
> > >
> > > |Method|Ratio|Arc-C|HellaS|Arc-E|Avg|
> > > |:-:|:-:|:-:|:-:|:-:|:-:|
> > > |ShortGPT|10|54.31|62.88|71.87|63.02|
> > > |IT-Prun||79.77|87.88|94.61|**87.42**|
> > > |ShortGPT|20|27.24|23.55|27.16|25.98|
> > > |IT-Prun||34.51|57.59|73.81|**55.30**|
> > > |ShortGPT|30|24.94|21.93|24.00|23.62|
> > > |IT-Prun||25.12|22.44|23.92|23.83|
> > >
> > > **[Pruned Layer Idx (Count from 0) of Llama3-8B]**
> > >
> > > |Prune Ratio|30%|
> > > |:-:|:-:|
> > > |Shortgpt|[24,23,25,26,27,28,22,21,29,19]|
> > > |IT-Prun|[24,23,25,26,27,28,22,21,19,20]|
> > >
> > > > **W4:  Results in Table 5**
> > >
> > > **We must state clearly that no test-set fine-tuning or data leakage occurred. We report the exact SFT/evaluation splits used in Table 5 below.**
> > >
> > > SFT/eval files: `BoolQ train/validation`; `ARC-Easy train/test`; `ARC-Challenge train/test`; `PIQA train/validation`; `HellaSwag train/validation`; `WinoGrande xl_train/xl_validation`. For MMLU, sft uses train when available, otherwise validation; eval uses test split.

---

### Official Review · Reviewer_CD6i · 2026-03-13

**Soundness:** 3
**Presentation:** 3
**Significance:** 3
**Originality:** 2
**Overall Recommendation:** 4
**Confidence:** 3

**Summary:**

The paper investigates why layer-pruned large language models often exhibit abrupt performance collapse. Focusing on multiple-choice tasks, it introduces a decision-centered framework with two metrics, Decision Margin (DM) and Option Frequency (OF), along with an iterative pruning method for tracking layer-wise decision dynamics. The key finding is a sharp transition from a “Silent Phase,” in which the model has not yet formed the correct prediction, to a “Decisive Phase,” in which the correct answer emerges. The results suggest that performance collapse occurs when pruning disrupts the Silent Phase and prevents this critical transition. Experiments on Llama2-7B, Llama3-8B, and Qwen3-4B, together with several ablation studies, support this conclusion.

**Compliance With Llm Reviewing Policy:**

Affirmed.

**Final Justification:**

The response addresses my main concerns adequately, especially by clarifying that Silent-phase disruption is presented as a dominant structural mechanism rather than the only possible cause, and showing that the transition-collapse pattern also appears under alternative pruning criteria and random controls. The additional stability analysis across subsets, prompt variants, and option-order perturbations also strengthens confidence that the main phenomenon is not a fragile artifact of a single evaluation setup.

At the same time, my overall assessment is not substantially changed. I still view the paper’s main contribution as primarily empirical and interpretive rather than methodological, and I continue to think the level of methodological novelty is somewhat limited. However, I do find the empirical phenomenon important, the phase-based framing useful, and the supporting analyses sufficiently thorough to make the paper a meaningful contribution.

Overall, the rebuttal resolves my main concerns, but does not fundamentally change my view of the paper’s strengths and limitations. I therefore maintain my score at Weak Accept.

**Key Questions For Authors:**

How stable is the estimated transition point across evaluation subsets, prompt variants, or random seeds?

**Limitations:**

yes

**Strengths And Weaknesses:**

**Strengths**
* The paper focuses on a concrete empirical phenomenon in layer-pruned large language models: sudden performance collapse. Understanding the origin of this behavior is important for developing more reliable pruning strategies and improving model interpretability.
* The empirical framing is intuitive and easy to understand. The Silent/Decisive phase decomposition provides a useful mental model for the observed cliff-edge behavior.
* The paper includes several complementary analyses rather than relying on a single metric. In particular, the combination of CKA, DM, OF/KL, noise sensitivity, and targeted ablations makes the study more substantial.

**Weaknesses**
* Limited methodological novelty. DM is essentially a margin-style statistic computed from logit-lens outputs, OF is a layer-wise argmax frequency measure, and IP mainly applies an existing BI-based criterion in a greedy iterative pruning loop. The main contribution lies more in the empirical findings and interpretation than in fundamentally new methodology.
* The paper argues that collapse stems from disruption of the Silent Phase, but the evidence is still largely observational, based on one pruning trajectory with a few targeted ablations. The current experiments suggest that Silent-Phase disruption is an important factor, but they do not rule out other mechanisms or establish it as the unique cause of collapse.
* The conclusions may be overly dependent on the BI-based iterative pruning procedure. Because the analysis follows the pruning trajectory induced by this heuristic, the observed transition-and-collapse pattern may partly be an artifact of the procedure itself. The paper should test whether the same pattern holds under alternative pruning criteria or random controls.

---

> ### Author Rebuttal · Authors · 2026-03-30
>
> Dear Reviewer CD6i,
>
> Thank you for your constructive comments. We address your concerns below:
>
> > **W1: Contribution and Novelty**
>
> Although DM, OF, and IP arenot individually new,  our core contribution is **a decision-centered perspective on pruning collapse**.  We identify a sharp **Silent/Decisive phase structure** and explain that  the **the failure to reach the decision transition** is a critical reason for performance collapse. We also prove that the transition layer and its neighbors act as a **structural bottleneck**, where protecting these layers significantly delays collapse (see Response to Reviewer RD6A on W1).
>
> > **W2: Mechanism behind Collapse**
>
> We clarify that Silent-phase disruption is a dominant structural mechanism, not the only cause of collapse. To further verify this, we add **sliding-window deletions** (3-layer windows) on Llama3-8B. Across tasks, removing layers from the Silent and Transition phases leads to much larger degradation than removing layers from the Decisive phase. These results support the importance of the Silent and Transition phases for preserving performance under layer pruning.
>
> |Llama3-8B|Early-Silent(0,1,2)|Mid-Silent(8,9,10)|Late-Silent(15,16,17)|Transition(16,17,18)|Early-Decisive(17,18,19)|Mid-Decisive(23,24,25)|Late-Decisive(29,30,31)|
> |:-:|:-:|:-:|:-:|:-:|:-:|:-:|:-:|
> |ARC-Challenge|0.232|0.510|0.346|0.364|0.836|0.830|0.840|
> |HellaSwag|0.232|0.270|0.450|0.412|0.510|0.716|0.744|
> |ARC-Easy|0.236|0.632|0.624|0.472|0.786|0.878|0.898|
>
> > **W3: Alternative pruning criteria**
>
> We compare IP (stepwise BI), ShortGPT (one-shot BI), MKA (manifold-based), and Random pruning methods. In the table, NT% is the pruning ratio where no transition is observed, CP% is the collapse ratio, Acc(C±10%) is pre/post-collapse accuracies, and Acc(C) is the accuracy at collapse. “—” means NaN due to coarse pruning granularity  (Dense, 10%, 20%, 30%, 40%, 50%).
>
> The results show that the same transition-collapse pattern is observed across all non-random methods considered here: accuracy remains stable until the transition disappears. **Random** pruning destroys the transition and causes collapse at much lower ratios (10%), **confirming that this pattern is not an artifact of BI-based iteration.**
>
> |Model|Method|NT%|CP%|Acc(C-10%)|Acc(C)|Acc(C+10%)|
> |:-:|:-:|:-:|:-:|:-:|:-:|:-:|
> |ARC-Challenge|IP|40|40|0.80|0.28|0.25|
> ||ShortGPT|40|40|0.47|0.23|0.21|
> ||MKA|50|50|0.80|0.27|—|
> ||Random|10|10|0.82|0.21|0.2|
> |HellaSwag|IP|40|40|0.64|0.28|0.26|
> ||ShortGPT|30|30|0.70|0.27|0.27|
> ||MKA|50|50|0.78|0.28|—|
> ||Random|10|10|0.76|0.20|0.22|
>
> > **Q1: The stability of evaluation subsets, prompt variants, or random seeds**
>
> We evaluate the stability of our findings on Llama3-8B / ARC-Challenge under three settings:
> (1) **evaluation subsets with 5 sampling seeds (2026–2030)**,
> (2) **3 prompt templates**, and
> (3) **2 seeds for option-order shuffling**.
> In the table, Acc (Mean/Std) denotes the mean and standard deviation of accuracy, T Exist Rate is the fraction of runs with the transition, T Idx (Mean/Std) denotes the mean and standard deviation of the transition layer index over runs where the transition exists, and Final DM (Mean/Std) refers to the mean and standard deviation of DM at the final layer.
>
> The same qualitative pattern holds across all three settings: in the **pre-collapse regime** (Dense–30%), the transition always exists and is consistently located at layer 18; in the **collapse regime** (40%–50%), the transition disappears and final-layer DM becomes consistently negative. This suggests that the estimated transition point is robust to subset sampling, prompt variation, and option-order perturbation.
>
> Due to space constraints, we report only **Llama3-8B / ARC-Challenge** here. The omitted results on **Qwen3-4B** and **ARC-Easy / HellaSwag** show the same qualitative pattern, and can be provided upon request.
>
> **[evaluation subsets]**
>
> |Pruning Ratio|Acc(Mean/Std)|T Exist Rate|T Idx(Mean/Std)|Final DM(Mean/Std)|
> |:-:|:-:|:-:|:-:|:-:|
> |Dense|0.83/0.0107|1|18/0|2.9228/0.0993|
> |10%|0.82/0.0084|1|18/0|1.9787/0.0655|
> |20%|0.77/0.0097|1|18/0|1.0112/0.0536|
> |30%|0.80/0.0099|1|18/0|2.0678/0.0927|
> |40%|0.28/0.0054|0|—|-0.7497/0.0123|
> |50%|0.24/0.0109|0|—|-1.0068/0.0387|
>
> **[prompt variants]**
>
> |Pruning Ratio|Acc(Mean/Std)|T Exist Rate|T Idx(Mean/Std)|Final DM(Mean/Std)|
> |:-:|:-:|:-:|:-:|:-:|
> |Dense|0.83/0.0059|1|18/0|2.7839/0.1482|
> |10%|0.82/0.0099|1|18/0|1.7659/0.1816|
> |20%|0.74/0.0224|1|18/0|0.8964/0.0695|
> |30%|0.78/0.0238|1|18/0|2.1089/0.1861|
> |40%|0.28/0.0047|0|—|-0.7708/0.0686|
> |50%|0.25/0.0127|0|—|-0.9760/0.0597|
>
> **[option-order perturbation seed]**
>
> |Pruning Ratio|Acc(Mean/Std)|T Exist Rate|T Idx(Mean/Std)|Final DM(Mean/Std)|
> |:-:|:-:|:-:|:-:|:-:|
> |Dense|0.838/0.0100|1|18/0|2.9861/0.0035|
> |10%|0.832/0.0020|1|18/0|2.1011/0.0798|
> |20%|0.793/0.0190|1|18/0|1.0551/0.0609|
> |30%|0.816/0.0020|1|18/0|2.1629/0.0981|
> |40%|0.270/0.0100|0|—|-0.7425/0.0113|
> |50%|0.230/0.0100|0|—|-1.0562/0.0052|

---

> > ### Author Rebuttal · Reviewer_CD6i · 2026-04-01
> >
> > Thank you for the response. I have no further questions.

---

> > > ### Author Response · Authors · 2026-04-03
> > >
> > > Thank you for the reviewer's kind Acknowledgement. We are pleased to hear that our previous explanation has answered your questions.

---

### Decision · Program_Chairs · 2026-04-30

**Decision:**

Accept (regular)

**Comment:**

Reviewers liked that the paper offers a clear, intuitive phase-based framework that explains pruning-induced collapse. All reviewers weakly advocate for acceptance. Accept.